# Multi-Armed Bandits with Interference:
# Bridging Causal Inference and Adversarial Bandits

**Su Jia** [1]  **Peter Frazier** [2]  **Nathan Kallus** [2]

## Abstract

Experimentation with interference poses a significant challenge in contemporary online platforms. Prior research on experimentation with interference has concentrated on the final output of a policy. Cumulative performance, while equally important, is less well understood. To address this gap, we introduce the problem of *Multi-armed Bandits with Interference* (MABI), where the learner assigns an arm to each of $N$ experimental units over $T$ rounds. The reward of each unit depends on the treatments of *all* units, and this dependence decays in distance. The reward functions, chosen by an adversary, may vary arbitrarily over space and time. We first show that a switchback policy achieves an optimal *expected* regret of $\tilde{O}(\sqrt{T})$ against the best fixed-arm policy; however, the regret as a random variable suffers high variance. We propose a policy based on a novel clustered randomization scheme, whose regret (i) is optimal in *expectation* and (ii) admits a high-probability bound that vanishes in $N$.

## 1. Introduction

A/B tests have become a standard practice to evaluate the impact of a new product or service change before wide-scale release. A naive A/B test may fail in the presence of **interference**, i.e., the *Stable Unit Treatment Values Assumption* (SUTVA, Rubin 1978) is violated, where the treatment of one unit affects the outcome of another. For example, if a ride-sharing firm assigns half of the drivers to a new pricing algorithm, these drivers will alter their behaviors, which impacts the common pool of passengers, and consequently the drivers not assigned the new algorithm.

Most previous work on experimentation with interference focused on the quality of the **final** output, such as the mean-squared error of the estimator (e.g., Ugander et al. 2013) or $p$-value (e.g., Athey et al. 2018). On the other hand, the **cumulative** performance is also important in practice, considering the scale of experiments on modern platforms, but this perspective is often overlooked.

This motivates us to study cumulative reward maximization in A/B testing with interference. We employ a batched adversarial bandits framework. Given a set $U \subseteq \mathbb{R}^2$ of $N$ *units* representing, for example, users in an online platform, each with a known, fixed location. We are also given a set $[k] := \{1, \ldots, k\}$ of *arms* and a time horizon with $T$ rounds. In each round, the learner assigns one arm to **each** unit and collects an observable reward, governed by a *reward function* secretly chosen by an adversary beforehand.

To capture **interference**, the mean reward of each unit depends on the treatments of **all** units. Formally, this means that each reward function is defined on $[k]^U$ instead of on $[k]$ (as in ordinary bandits). Similarly to causal inference with interference, efficient learning is impossible without additional structures. In this work, we employ a rather general assumption from Leung 2022 that the unit-to-unit interference decreases in their distance. Specifically, for any two treatment assignments $z, z' \in [k]^N$, if they are identical on a neighborhood of $u \in U$, then their rewards on $u$ are close; Moreover, the larger this neighborhood, the closer they are.

On the **technical** level, we build upon the EXP3 framework for adversarial bandits, integrating with a novel ***H**orvitz-**T**hompson estimator* with **I**mplicit e**X**ploration (HT-IX). By choosing different hyperparameters, this estimator "interpolates" between the estimator for the *average treatment effect* (ATE) under spatial interference (Leung, 2022) and for the unobserved rewards in adversarial bandits (Kocák et al., 2014; Neu, 2015).

From a practical standpoint, a "pure" switchback policy — where the entire system switches between treatment and control — is rarely used in practice. Instead, platforms typically partition the space into clusters and independently assign treatment or control to each cluster in each period; see, e.g., DoorDash's "clustered" switchback experiments

[1]Center for Data Science for Enterprise and Society, Cornell University [2]School of Operations Research and Information Engineering, Cornell University. Correspondence to: Su Jia <jiasu1225@gmail.com>.

*Proceedings of the $42^{nd}$ International Conference on Machine Learning*, Vancouver, Canada. PMLR 267, 2025. Copyright 2025 by the author(s).

(Sneider and Tang, 2019). This is precisely the class of policies that we analyze in this work. Thus, a key contribution of our work is to provide **theoretical justification** for a policy class widely used in industry.

### 1.1. Our Contributions

We contribute to the literature in the following ways.

**1. Bridging Adversarial Bandits and Causal Inference.** We incorporate interference into MAB by formulating the problem of *Multi-Armed Bandits with Interference* (MABI). The ordinary MAB can therefore be seen as a special case of MABI under SUTVA. Our formulation is fairly general, imposing no constraints on the non-stationarity or heterogeneity in reward functions between units. At the heart of our policy is an estimator that generalizes the estimators of (1) Kocák et al. (2014) for deriving high-probability regret bound in adversarial bandits and (2) Leung (2022) for estimating treatment effects under spatial interference.

**2. Optimal Expected Regret.** We show that a switchback policy has an optimal (up to log terms) $\widetilde{O}(\sqrt{kT})$ expected regret. Moreover, we also show that for any (possibly non-switchback) policy, there is a MABI instance on which it suffers an $\Omega(\sqrt{kT})$ expected regret. Notably, this suggests that a large $N$ does not help reduce the **expected** regret.

**3. High-probability Bound.** Although the regret (as a random variable) of a switchback policy may be optimal in expectation, it can suffer high variance. To address this, we propose a policy that integrates the following components.
**a) Randomized Clustered Randomization.** We introduce the *Robust Random Partition* (RRP) where we perturb the cluster boundaries randomly. This increases the exposure probability from $p^{O(1)}$ to $\Omega(p)$. This reduces the variance caused by the specific clustering chosen for cluster randomization.
**b) Estimator.** Our HT-IX estimator adds an *implicit exploration* (IX) parameter (Kocák et al., 2014) into the propensity score of the Horvitz-Thompson (HT) estimator. This reduces the variance caused by unbalanced weights over the treatment arms.

We show that the EXP3 policy based on (1) the RRP design and (2) the HT-IX estimator has an optimal expected regret. Moreover, the tail mass of the regret vanishes as $N \to \infty$. In stark contrast, this result is **not** possible for any switchback policy, as the tail mass of the regret does **not** depend on $N$. This result is crucial for practical applications, as market size $N$ is typically orders of magnitude larger than $T$.

### 1.2. Related Work

Experimentation is a widely deployed learning tool in online commerce that is easy to execute (Kohavi and Thomke, 2017; Thomke, 2020; Larsen et al., 2023). As a key chal-

lenge, the violation of the SUTVA has been viewed as problematic for online platforms (Blake and Coey, 2014). This problem has been extensively studied in statistics (e.g., Hudgens and Halloran 2008; Aronow and Samii 2017; Eckles et al. 2017; Basse and Feller 2018; Basse et al. 2019; Li and Wager 2022; Hu et al. 2022; Leung 2023; Hu and Wager 2022), operations research (e.g., Johari et al. 2022; Bojinov et al. 2023; Farias et al. 2022; Holtz et al. 2024; Candogan et al. 2024; Jia et al. 2023a), computer science (e.g., Ugander et al. 2013; Saveski et al. 2017; Ugander and Yin 2023; Yuan et al. 2021) and medical research (Tchetgen and VanderWeele, 2012). Some recent surveys include Bajari et al. 2023; Larsen et al. 2023.

Many works tackle this problem by assuming that interference is summarized by a low-dimensional exposure mapping and that units are individually randomized to treatment or control by Bernoulli or complete randomization (Manski, 2013; Toulis and Kao, 2013; Aronow and Samii, 2017; Basse et al., 2019; Forastiere et al., 2021). To improve estimator precision, some work departed from unit-level randomization and introduced cluster correlation in treatment assignments. This is usually done by either (i) grouping the units in a network into clusters (Ugander et al., 2013; Jagadeesan et al., 2020; Leung, 2022; 2023) or (ii) grouping time periods into blocks ("switchback") (Bojinov et al., 2023; Hu and Wager, 2022; Jia et al., 2023a). However, these works usually focus on the quality of the final output, such as the bias and variance of the estimator (Ugander et al., 2013; Leung, 2022) and $p$-values for hypothesis testing (Athey et al., 2018).

While existing literature has primarily focused on the final output, the cumulative performance remains less well understood. A natural framework is *multi-armed bandits* (MAB) (Lai et al., 1985). These works focus on the cumulative performance but often overlook the interference element. There are three lines of work in MAB that are most related to this work: (i) adversarial bandits, (ii) multiple-play bandits and (iii) combinatorial bandits.

Particularly related is the adversarial bandit problem. Many policies for adversarial bandits are built on the idea of weight update (Vovk, 1990; Littlestone and Warmuth, 1994), first introduced for the full-information setting (i.e., the *best expert problem*). Auer et al. (1995) considered the bandit feedback version and proposed a forced exploration version of the EXP3 policy. Stoltz (2005) observed that the policy achieves the optimal expected regret even without additional exploration.

High-probability bounds for adversarial bandits were first provided by Auer et al. (2002) and explored in a more generic way by Abernethy and Rakhlin (2009). In particular, the idea to reduce the variance of importance-weighted estimators has been applied in various forms (Ionides, 2008;

Bottou et al., 2013) and was first introduced to bandits by Kocák et al. (2014). Subsequently, Neu (2015) showed that this algorithm admits high-probability bounds.

Another closely related line is *multiple-play bandits* (Anantharam et al., 1987), where the learner plays multiple arms per round and observes **each** of their feedback. The number of arms played in each round can be viewed as the "$N$" in our problem (Chen et al., 2013; Komiyama et al., 2015; Lagrée et al., 2016; Jia et al., 2023b). Another related line of work is multi-agent RL (Kanade et al., 2012; Busoniu et al., 2008; Zhang et al., 2021). Our problem differs in that each "agent" behaves completely passively.

Finally, since the reward function is defined on the hypercube $[k]^N$, our work is also related to *combinatorial bandits* (Cesa-Bianchi and Lugosi, 2012) where the action set is a subset of a binary hypercube. While most work in this area considers linear reward functions, in our work the reward functions are only assumed to satisfy the decaying interference assumption. A recent line of work focuses on combinatorial bandits with non-linear reward functions. However, most of these works either assume a stochastic setting (Agrawal et al., 2017; Kveton et al., 2015) or an adversarial setting with a restrictive class of reward functions, such as polynomial link functions (Han et al., 2021).

**Most closely** related is the concurrent work of Agarwal et al. (2024); Zhang and Wang (2024); Xu et al. (2024) on **stochastic** bandits with interference. Specifically, the reward at each unit is determined by a **stationary** reward function. With stationarity, they can adopt a stronger benchmark, e.g., Agarwal et al. (2024) chose the best "personalized" treatment assignment, as opposed to the best **uniform** treatment assignment in our work. However, our framework is **more general** in two ways: (1) the reward functions can be non-stationary heterogeneous arbitrarily, and (2) we do not impose much structural assumptions (such as linearity) on the interference pattern; we only assume that the interference level decays in distance.

## 2. Formulation and Assumptions

We consider a *multiple-play* (i.e., multiple arms are played in each round) adversarial bandit setting. Consider a set of $N$ units, $T$ *rounds* and $k$ *treatment arms* (or *arms*). For each round $t \in [T]$ and unit $u \in [N]$ there is an unknown *reward function* $Y_{ut} : [k]^N \to [0,1]$ that can depend on the *treatment assignment* $z \in [k]^N$ (not just $z_u$).

In each round $t$, the learner selects a *treatment assignment* $Z_t \in [k]^N$ and observes a reward $Y_{ut}(Z_t)$ for each $u \in [N]$ (i.e., *bandit feedback*). Specifically, the distributions we use to draw $(Z_t)$ form a *policy* which is, formally, a sequence $\pi_t : ([k]^N \times [0,1]^N)^{t-1} \to \Delta([k]^N)$ where $t = 1, \ldots, T$.

As in adversarial bandits, we aim to control the loss compared to the best fixed arm. When $N = 1$, our notion of regret is the same as in adversarial bandits.

**Definition 2.1** (Regret). The *regret* of a policy $Z$ is defined as $\text{Reg}(Z) := \max_{a \in [k]} \{\text{Reg}(Z, a)\}$ where for each $a \in [k]$, we define

$$\text{Reg}(Z, a) := \sum_{t=1}^{T} \frac{1}{N} \sum_{u \in [N]} (Y_{ut}(a \cdot \mathbf{1}^N) - Y_{ut}(Z_t))$$

We focus on bounding the regret in expectation or, preferably, in high probability. The reward functions are chosen "secretly" in advance in that our bounds on regret will hold for **any** reward functions (possibly subject to some constraints we discuss next). Thus, the bounds hold even for the worst-case reward functions chosen by an adversary.

Thus far, the model allows for **unrestricted interference** in that $Y_{ut}(z)$ may vary arbitrarily in any coordinate of $z$. As in the literature of experimentation under interference, to derive meaningful performance guarantees, it is necessary to assume certain structures on interference. The existing literature focuses on the restrictions captured by $\kappa$-neighborhood exposure mappings, which imply that the arm assigned to $v$ (i.e., $z_v$) can only interfere with $Y_{ut}$ if the distance between $u, v$ is at most $\kappa$, which is quite restrictive.

In many applications, the effect of a treatment diffuses primarily through physical interaction, such as a promotion on a ride-sharing platform or a discount on a food delivery platform. Leung (2022) addressed this by proposing a model that allows for interference between any two units, with an intensity that **decays** in the distance. The following is identical to that in their §2.1 (up to re-scaling).

**Assumption 2.2** (Scaling of the Bounding Box, Leung 2022). There is $b_N = O(\sqrt{N})$ s.t. $U \subseteq [-b_N, b_N]$ and $d(u, v) \geq 1$ for any $u, v \in [N]$ where $d$ is the sup norm.

Leung (2022) posits that if two assignments $z, z'$ are identical on a ball-neighborhood of $u$, then the mean rewards of $u$ under $z, z'$ are close. To formalize, denote the radius-$r$ (open) *ball* as $B(u, r) := \{v \in [N] \mid d(u, v) < r\}$. For clarity, $B(u, r)$ does not contain units that are **exactly** distance $r$ away. In particular, $B(u, 0) = \emptyset$ and $B(u, 1) = \{u\}$.

**Definition 2.3** (Decaying Interference Property). Let $\psi : [0, \infty) \to [0, \infty)$ be non-increasing. A MABI instance satisfies the $\psi$-*decaying interference property* (or $\psi$-*DIP*) if for any $r \geq 0, u \in [N], t \in [T]$ and $z, z' \in [k]^N$ with $z_{B(u,r)} = z'_{B(u,r)}$, we have

$$|Y_{ut}(z) - Y_{ut}(z')| \leq \psi(r).$$

*Remark* 2.4 (Recovering SUTVA). Consider $\psi(r) = \mathbb{1}(r = 0)$. Then, for any $z, z'$ identical on $B(u, 1) = \{u\}$ and $r >$

0 in Definition 2.3, we have $|Y_{ut}(z) - Y_{ut}(z')| \leq \psi(1) = 0$. Thus, $Y_{ut}(z)$ only depends on $z_u$, which recovers SUTVA.

# 3. Expected Regret

We show that the minimax *expected* regret is $\widetilde{O}(\sqrt{T})$. This holds for all $N$ (**not fixed as constant!**), although $N$ does not show up explicitly. This is not surprising due to the normalization factor $1/N$ in the definition of regret.

## 3.1. Upper Bound

We begin by observing that MABI is equivalent to adversarial bandits when we restrict ourselves to switchback policies, which selects the same arm for all units in each round. These policies are widely applied in practice (Sneider and Tang, 2019; Cooprider and Nassiri, 2023) and have been extensively studied (Bojinov et al., 2023; Hu and Wager, 2022; Xiong et al., 2023). Note that any adversarial bandit policy $(A_t)$ induces a switchback policy $(Z_t)$ where $Z_t = A_t \cdot \mathbf{1}^N$. Moreover, this reduction preserves the regret:

**Proposition 3.1** (Reduction to Adversarial Bandits). *Let $(A_t)$ be an adversarial bandits policy with expected regret $r(T)$. Then, the induced switchback policy $(Z_t)$ satisfies $\mathrm{Reg}(Z) = r(T)$.*

For adversarial bandits, the EXP3 ("EXPlore and EXPloit with EXPonential weights") policy has an $\widetilde{O}(\sqrt{kT})$ expected regret (Auer et al., 1995), so:

**Corollary 3.2.** *The EXP3-based switchback policy has expected regret $\widetilde{O}(\sqrt{kT})$.*

## 3.2. Lower Bound on the Expected Regret

The meticulous reader may have noticed that Corollary 3.2 does **not** involve the market size $N$. This is because switchback policies treat the entire system as a whole, and do not involve $N$. Can we improve the bound by leveraging $N$ using more complicated policies? The answer is **no**:

**Theorem 3.3** (Lower Bound on the Expected Regret). *Fix any non-increasing function $\psi$ with $\psi(0) = 1$ and $\lim_{x \to \infty} \psi(x) = 0$. Then for any MABI policy $Z$, there exists a MABI instance $\mathcal{I}$ satisfying the $\psi$-DIP and Assumption 2.2 s.t. $\mathrm{Reg}(Z, \mathcal{I}) = \Omega(\sqrt{kT})$.*

We will choose the units to be integer grid points $U = \{-2\sqrt{N}, 2\sqrt{N}\} \times \{-2\sqrt{N}, 2\sqrt{N}\}$, which obviously satisfies Assumption 2.2. To highlight key ideas, let us assume $k = 2$. The extension to general $k$ is straightforward.

**High Level Idea.** We will choose the $k \times T$ reward table as a random matrix with i.i.d. Bernoulli entries with means $1/2$. We then argue that
(a) the expected (over the randomness of the rewards and

policy) regret of any policy is $T/2$, and
(b) by a standard anti-concentration bound, w.h.p. there is a fixed-arm policy with total reward $T/2 + \Omega(\sqrt{T})$.

**Challenge.** At first sight, the proof seems to follow **trivially** from the lower bound of the best-expert problem (see, e.g., Section 4 of Arora et al. 2012), by assigning all units the same arm in each round. However, by doing so, the reward functions are defined only for $\mathbf{1}^N$ and $\mathbf{0}^N$, but we need to specify their values on the entire $\{0, 1\}^N$, subject to the $\psi$-DIP. As a **key step**, we show that such an extension is always possible:

**Lemma 3.4** (Hypercube Extension). *Let $G = (V, E)$ the grid graph where $V = \{-m, \ldots, m\} \times \{-m, \ldots, m\}$ for some integer $m$. Then, for any non-increasing $\psi : \mathbb{R}_+ \to \mathbb{R}_+$, there is a function $f : \{0, 1\}^V \to [0, 1]$ satisfying*
**i) the boundary condition:** *$f(\mathbf{0}^V) = 0$, and $f(\mathbf{1}^V) = \psi(0) - \psi(m)$, and*
**ii) the $\psi$-DIP:** *for any $z, z' \in \{0, 1\}^V$, if $z_{B(O,r)} = z'_{B(O,r)}$ for some $r > 0$ (where $O = (0, 0)$), then $|f(z) - f(z')| \leq \psi(r)$.*

*Proof.* We construct $f$ as follows.
**Step 1: Define $f$ on Basis Vectors.** For each $r = 0, \ldots, m$, we define the *basis vector* $\boldsymbol{\sigma}^r \in \{0, 1\}^V$ with entries

$$\sigma_v^r = \mathbb{1}(\|O - v\|_\infty < r), \forall v \in V.$$

In particular, $\boldsymbol{\sigma}^0 = \mathbf{0}^N$ and $\boldsymbol{\sigma}^m = \mathbf{1}^N$. Define

$$f(\boldsymbol{\sigma}^r) := \psi(0) - \psi(r), \quad r = 0, \ldots, m.$$

**Step 2: Extend $f$ to $\{0, 1\}^V$.** For each $z \in \{0, 1\}^V$, define $f(z) = f(\boldsymbol{\sigma}^{r_\star(z)})$ where

$$r_\star(z) = \max\{r \geq 0 : z_{B(O,r)} = \mathbf{1}_{B(O,r)}\}.$$

Note that $f(\boldsymbol{\sigma}^0) = \psi(0) - \psi(0) = 0$ and $f(\boldsymbol{\sigma}^m) = \psi(0) - \psi(m)$, and so (i) holds. To show (ii), fix any $z, z' \in \{0, 1\}^V$. For simplicity, we write $r_\star := r_\star(z)$ and $r'_\star = r_\star(z')$. W.l.o.g. we assume that $r_\star \leq r'_\star$. Consider the largest ball $B(O, \rho)$ on which $z$ and $z'$ are identical, that is,

$$\rho := \max\{r \geq 0 : z_{B(O,r)} = z'_{B(O,r)}\}.$$

*Claim* 3.5. $\rho \geq r_\star$.

Assuming the claim (whose proof is deferred to Appendix B), we conclude that

$$
\begin{aligned}
|f(z) - f(z')| &= |(\psi(0) - \psi(r_\star)) - (\psi(0) - \psi(r'_\star))| \\
&= \psi(r_\star) - \psi(r'_\star) \\
&\leq \psi(r_\star) \leq \psi(\rho),
\end{aligned}
$$

where the last inequality is because $\psi$ is non-increasing. $\square$

**Proof Sketch of Theorem 3.3.** Recall that $U = \{-2\sqrt{N}, 2\sqrt{N}\} \times \{-2\sqrt{N}, 2\sqrt{N}\}$.

**Step 1: Hypercube Extension.** W.l.o,g, we assume $\psi(0) = 1$ and $\psi(\sqrt{N}) = 0$. Consider the interior units

$$U_{\text{int}} := \{-\sqrt{N}, \sqrt{N}\} \times \{-\sqrt{N}, \sqrt{N}\}.$$

For each $u \in N \backslash U_{\text{int}}$, we set $Y_{ut} \equiv 0$. For each $u = (u_x, u_y) \in N_{\text{int}}$, apply Lemma 3.4 with $m = \sqrt{N}$ and

$$V = \{u_x - m, u_x + m\} \times \{u_y - m, u_y + m\}.$$

Denote by $\tilde{f}_u : \{0, 1\}^V \to \mathbb{R}_+$ the function constructed. Extend $\tilde{f}_u$ to $\{0, 1\}^N$ so that the function value is solely determined by its restriction on $V$.

**Step 3: Construct the reward function.** Consider i.i.d. $\xi_t \sim \text{Ber}(1/2)$ for $t \in [T]$. Define the reward function $Y_{ut}(z) := \frac{1}{2} + \left(\xi_t - \frac{1}{2}\right) f_u(z)$ where $z \in \{0, 1\}^N$. By Lemma 3.4, $Y_{ut}(\cdot)$ satisfies the $\psi$-DIP. Moreover, $\mathbb{E}_\xi[Y_{ut}(z)] = \frac{1}{2}$ for any $z \in \{0, 1\}^N$, and so the expected reward of **any** policy is $T/2$.

**Step 4: Applying anti-concentration bound.** By a standard anti-concentration bound, the best fixed arm has a total expected reward of $T/2 + \Omega(\sqrt{T})$. It then follows that

$$\mathbb{E}\left[\max\{R_0, R_1\} - \frac{T}{2}\right] = \Omega\left(\sqrt{T}\right). \qquad \square$$

# 4. High Probability Regret Bound

Although switchback policies can achieve optimal expected regret, their disregard for $N$ prevents them from capitalizing on the market size. Consequently, the variance of regret (as a random variable) does not vanish as $N$ grows, making these policies less appealing to practitioners. To address this, we propose a policy that (i) has a $\widetilde{O}(\sqrt{T})$ expected regret and (ii) admits a h.p.-bound that vanishes in $N$. Specifically, for any $T$ and confidence level $\delta > 0$, the tail mass above $\widetilde{\Theta}(\sqrt{T})$ vanishes $N \to \infty$. Our policy integrates EXP3-IX from adversarial bandits and the idea of clustered randomization from causal inference.

## 4.1. Background and Technical Challenges

The **first key idea** is implicit exploration in adversarial bandits. Policies for adversarial bandits often rely on *weights update* (Vovk, 1990; Littlestone and Warmuth, 1994). In each round, an arm is selected with a probability proportional to its weight, updated to incentivize choosing arms with historically high rewards.

However, with bandit feedback, we only observe the reward of the selected arm. EXP3 addresses this using an importance-weighted estimator for the rewards of *all* arms (Auer et al., 1995). In round $t$, let $P_{ta}$ be the probability of selecting $a$, $A_t$ be the (random) arm selected, and $Y_t$ be its observed reward, then the estimate is

$$\widehat{Y}_{ta} := \frac{\mathbb{1}(A_t = a)}{P_{ta}} Y_t.$$

EXP3 combines this estimator with the multiplicative weights algorithm, and achieves an optimal $\widetilde{O}(\sqrt{kT})$ expected regret against the best fixed arm.

However, the regret has a **high variance**, potentially being *linear* in $T$ w.p. $\Omega(1)$;[1] see Note 1 in Chapter 11 of Lattimore and Szepesvári 2020. This is because the estimator in EXP3 can have a high variance. To address this, Kocák et al. (2014) introduced an *implicit exploration* ("IX") term $\beta > 0$ in the propensity weight, which truncates the value of the estimator and reduces its variance. Formally,

$$\widehat{Y}_{ta} := \frac{\mathbb{1}(A_t = a)}{P_{ta} + \beta} Y_t.$$

Despite the extra bias, Neu (2015) showed that judicious selection of $\beta$ leads to a good h.p. bound: The regret is $\lesssim \sqrt{\log 1/\delta}$ times the minimax regret, $\widetilde{O}(\sqrt{kT})$, w.p. $1-\delta$.

The other **key idea** is *clustered randomization*. Leung (2022) considered a uniform partition of the plane into square clusters, and independently assigned arms to each cluster. Under this design, the truncated HT estimator achieves a favorable bias-variance tradeoff, both vanishing in $N$.

There are two main **challenges** in integrating the truncated HT estimator into the EXP3-IX framework. First, the uniform spatial clustering in Leung 2022 is not "robust" since some arms may have very low probabilities due to weight updates, leading to high variance in the estimator. Furthermore, it is unclear how to select the IX parameter in the batched setting due to heterogeneity across units. For example, units lying close to the boundary of a cluster should have different IX parameters compared to those in the "interior".

The rest of this section focuses on addressing these challenges. The key component of the HT estimator in Leung 2022 is the exposure mapping, which measures the "reliability" of the data observed from a unit.

**Definition 4.1** (Exposure Mapping). We define the *radius-$r$ exposure mapping* as $X_{uta}^r(z) := \mathbb{1}(z_{B(u,r)} = a \cdot \mathbf{1}_{B(u,r)})$. For a random vector $Z \in [k]^N$, the *exposure probability* is $Q_{uta}^r(Z) := \mathbb{P}[X_{uta}^r(Z) = 1]$.

To control the variance, we prefer a high exposure probability. However, the exposure probabilities in a naive clustered randomization can be very low. We address this next.

---

[1] This does not contradict the (expected) regret bound, since the regret against the best fixed arm can be negative.

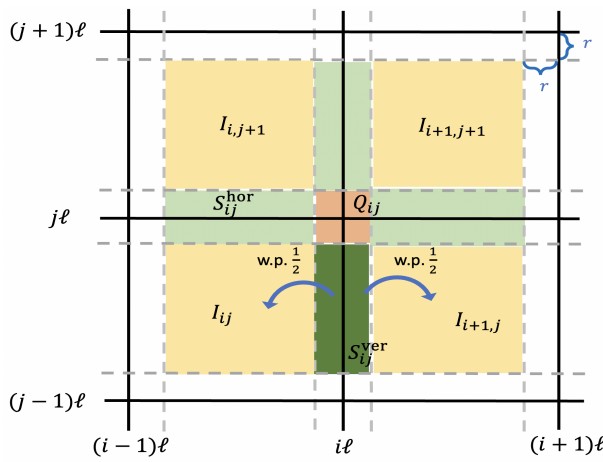

*Figure 1.* **Illustration of the RRP.** The black lines are the cluster boundary in the uniform clustering. We color the strips and quads green and red. We assign each strip to one of the two neighboring clusters; see $S_{ij}^{\text{ver}}$ (dark green). Finally, assign each quad (red) to one of the four nearby clusters with equal probabilities.

### 4.2. Robust Random Partition

If $T = 1$ and we aim to estimate the ATE, uniform clustering works well. For example, in Leung 2022, by choosing $r < \ell/2$ where $\ell$ is the side length of the square clusters, each $r$-ball can intersect only 4 squares. Therefore, if we independently assign each cluster an arm $a$ w.p. $p = \Omega(1)$ (and some other arm w.p. $1 - p$), the exposure probability is $p^4 = \Omega(1)$, which is favorable for estimation.

However, with **weight updates**, $p^4$ can be tiny. When this occurs, we almost remove all the data from units close to the cluster boundary, resulting in a high bias. For example, if the rewards are zero everywhere except near the boundaries, then our estimate is 0 while the true reward is $\Omega(1)$.

We address this by introducing **randomness** into the partition. We start with uniform clustering and then randomly assign units close to the boundary to nearby clusters. To formalize, recall from Assumption 2.2 that $U \subseteq [0, b_N]^2$ where $b_N = O(\sqrt{N})$. For any $\ell$ and $r < \ell/2$, an $(\ell, r)$-*robust random partition* (RRP) $\Pi = \{C_{ij} : 1 \leq i, j \leq b_N/\ell\}$ is defined as follows:

**1. Assign the Interiors:** Define the $(i, j)$-*interior* as

$$I_{ij} = [(i-1)\ell + r, \; i\ell - r] \times [(j-1)\ell + r, \; j\ell - r].$$

We assign $I_{ij}$ to $C_{ij}$ w.p. 1.
**2. Assign the Strips:** Define the *vertical* $(i, j)$-*strip*

$$S_{ij}^{\text{ver}} = [i\ell - r, \; i\ell + r] \times [(j-1)\ell + r, \; j\ell - r],$$

and the *horizontal* $(i, j)$-*strip*

$$S_{ij}^{\text{hor}} = [(i-1)\ell + r, \; i\ell - r] \times [j\ell - r, \; j\ell + r].$$

Assign $S_{ij}^{\text{ver}}$ independently to $C_{ij}, C_{i+1,j}$ uniformly. Similarly, assign $S_{ij}^{\text{hor}}$ to independently to $C_{ij}, C_{i,j+1}$ uniformly.
**3. Assign the Quads:** Define the $(i, j)$-*quad*

$$Q_{ij} = [i\ell - r, i\ell + r] \times [j\ell - r, j\ell + r].$$

Assign it to $C_{ij}, C_{i+1,j}, C_{i,j+1}, C_{i+1,j+1}$ uniformly. $\qquad\square$

Our clustering is obtained by partitioning $U$ using an RRP. Formally, let $\{C_{ij}\}$ be an $(\ell, r)$-RRP of $[0, \sqrt{N}]^2$. By abuse of notation, write $[N] \cap C_{ij}$ as $C_{ij}$. We will use the clustering $\{C_{ij} \mid 1 \leq i, j \leq b_N/\ell\}$. For each $u \in [N]$, denote by $C[u] \subseteq U$ the unique cluster that contains $u$. From now on, let us fix a pair of $\ell, r \geq 0$ with $1 \leq \ell \leq b_N$ and $2r < \ell$. The RRP enjoys the following nice *robustness*.

**Proposition 4.2** (Robustness). $\mathbb{P}[B(u, r) \subseteq C[u]] = \Omega(1)$.

To see this, observe that since $r < \ell/2$, the ball $B(u, r)$ intersects at most 4 "regions" (i.e., interiors, strips or quads). Since each strip or quad is assigned to a cluster independently, w.p. $\Omega(1)$ these regions are all assigned to the same cluster $C$. When this occurs, we have $B(u, r) \subseteq C[u]$.

The robustness boosts the exposure probability. In fact, our policy (to be defined soon) maintains a weight for each arm and assigns a random arm to each cluster independently according to the weights. Crucially, when $P_{ta}$ is small, the exposure probability under our RRP is $\Omega(P_{ta})$, which is much greater than the exposure probability $(P_{ta})^4$ under the uniform design (Leung, 2022). We will soon see how this helps reduce the variance of our estimator.

*Remark* 4.3. Our approach may resemble that of (Ugander and Yin, 2023). Their **randomized graph clustered randomization** (RGCR). While their randomized partitioning works for an arbitrary graph, our approach explicitly requires embedding the units into a Euclidean space. However, this structure makes it easier to exploit the decay of interference with distance. $\qquad\square$

### 4.3. HT-IX Estimator and Our Policy

In adversarial bandits, the key to designing a good policy is estimating the **unobserved** reward. More explicitly, we need to find a good estimator for the mean rewards $\{\bar{Y}_t(a \cdot \mathbf{1}^N)\}_{a \in [k]}$, where we recall that $\bar{Y}_t(Z) = \frac{1}{N} \sum_{u \in [N]} Y_{ut}(Z)$. Ideally, a good estimator enjoys both low bias and low variance. Neu (2015) showed that incorporating an additional IX parameter into the propensity weight leads to a favorable h.p. regret bound. In our MABI problem, the propensity weights $Q_{uta}^r$ can vary across units. However, a uniform IX parameter suffices for our result.

**Definition 4.4** (Horvitz-Thompson-IX Estimator). Fix an IX parameter $\beta \in [0, \frac{1}{2})$. Suppose the design

**Algorithm 1** EXP3-HT-IX Policy

1: Input:
   $\eta \in (0, 1)$: learning rate,
   $\beta \in [0, \frac{1}{2})$: IX parameter for the HT-IX estimator,
   $(\ell, r)$: parameters for the RRP.
2: $W_{ta} \leftarrow 1$ for each $a \in [k]$
3: **for** $t = 1, \dots, T$ **do**
4:     $W_t \leftarrow \sum_{a \in [k]} W_{ta}$                    *// Total weights*
5:     For each arm $a \in [k]$, let $P_{ta} \leftarrow W_{ta}/W_t$
6:     Randomly generate $\Pi_t$, an $(\ell, r)$-RRP
7:     **for** cluster $C \in \Pi_t$ **do**
8:         Draw an arm $Z_{Ct}$ using $(P_{ta})_{a \in [k]}$
9:         **for** $u \in C$ **do**
10:             $Z_{ut} \leftarrow Z_{Ct}$   *// Assign arm $A_t$ to all units in C*
11:         **end for**
12:     **end for**
13:     Observe the rewards $\{Y_{ut}(Z_t)\}_{u \in [N]}$
14:     **for** $a \in [k]$ **do**
15:         $\widehat{Y}_t(a) \leftarrow \frac{1}{N} \sum_{u \in [N]} \frac{\mathbb{1}(X^r_{uta}(Z_t)=1)}{Q^r_{uta}+\beta} Y_{ut}(Z_t)$
16:         $W_{t+1,a} \leftarrow e^{\eta \widehat{Y}_t(a)} W_{ta}$              *// Weight update*
17:     **end for**
18: **end for**

---

$Z_t \in [k]^N$ is drawn from a distribution $\mathcal{D}$.[2] Denote $Q^r_{uta} := \mathbb{P}_{Z_t \sim \mathcal{D}}(X^r_{uta}(Z_t) = 1)$. For any $t, a$, the *Horvitz-Thompson-IX* (HT-IX) estimator is

$$\widehat{Y}_t(a) := \frac{1}{N} \sum_{u \in [N]} \frac{\mathbb{1}(X^r_{uta}(Z_t) = 1)}{Q^r_{uta} + \beta} Y_{ut}(Z_t).$$

*Remark* 4.5 (Unifying Known Estimators). When $\beta = 0$, HT-IX becomes the HT estimator in Leung 2022; When $N = 1$, it becomes the estimator in EXP3-IX of Neu 2015.

We now **informally describe** our policy (Algorithm 1). It involves two parameters: The learning rate $\eta \in (0, 1)$, which controls how quickly we discount past data, and the IX parameter $\beta \in [0, \frac{1}{2})$ which truncates the HT-IX estimator by $1/\beta$. In each round, we **independently** generate an $(\ell, r)$-RRP. Then, we randomly assign an arm to each cluster independently, using the distribution determined by the weights. Finally, for each arm, we use the HT-IX estimator to estimate the counterfactual reward that we could have earned if we assigned it to **all** units in this round. We update the arm weights using the estimated rewards.

---

[2]More concretely, we will later choose $\mathcal{D}$ to be a clustered-level randomization, that is, assign a random arm to each cluster drawn with a probability proportional to its weight.

## 4.4. A High-probability Regret Bound

Denote by $R = \sum_{t=1}^T \frac{1}{N} \sum_{u \in [N]} Y_{ut}(Z_t)$ the reward of a policy $(Z_t)$. To compare against a fixed arm, also define

$$R_a := \sum_{t=1}^T \bar{Y}_t(a)$$

where

$$\bar{Y}_t(a) := \frac{1}{N} \sum_{u \in [N]} Y_{ut}(a \cdot \mathbf{1}^N).$$

**Theorem 4.6** (High-probability Bound). *Fix any IX parameter $\beta \in (0, \frac{1}{2})$ and learning rate $\eta \in (0, 1)$ in Algorithm 1. Then, for any $a^* \in [k]$ and $\delta \in (0, 1)$, we have $R - R_{a^*} = A + B + C$ w.p. $1 - \delta$ where*

$$A \lesssim \frac{\log k}{\eta} + \eta kT, \quad B \lesssim \left(\frac{1}{\beta} + \eta kT\right) \frac{\ell^2}{N} \log \frac{1}{\delta}$$

*and*

$$C \lesssim \beta kT + \frac{rT}{\ell} + \eta T \frac{k}{\beta} \psi(r)^2 + \frac{T}{\beta \ell^2} \psi(r).$$

We will soon see that with suitable parameters:

- $A$ becomes the minimax expected regret $\tilde{O}(\sqrt{kT})$,

- $B$ bounds the tail mass of the regret which vanishes as $N \to \infty$, and

- $C$ bounds the lower order terms.

We illustrate the importance of Theorem 4.6 via several corollaries. To minimize $A$, choose $\eta = \sqrt{\log k / kT}$. The choice for $\beta$ is more involved. With some foresight, let us choose $\beta = \sqrt{\frac{\ell^2}{kNT} \log \frac{1}{\delta}}$. To highlight the excess beyond the "necessary" regret, we denote $\text{Reg}_{\text{OPT}} := \sqrt{kT \log k}$ as the order-optimal expected regret.

We first consider the no-interference setting. In this case, our problem is equivalent to the multi-play (i.e., play $N$ arms in each round) variant of adversarial bandits. Since there is no interference, we will choose the singleton clustering (i.e., where each unit alone is a cluster).

This clustering can be realized as an $(\ell, r)$-RRP for suitable $\ell$ and $r$. In fact, recall from Assumption 2.2 that $d(u, v) \geq 1$ for any $u, v \in [N]$. Therefore, if we partition the bounding box $[0, b]^2$ uniformly into squares of sufficiently small side lengths $\ell$, then each square contains at most one unit in $U$. Thus, the $(\ell, r)$-RRP is just the singleton clustering (whenever $r < \ell/2$). Finally, noting that $\psi(r) = 0$ for any $r > 0$ and hence $C \lesssim \beta kT + o(1)$ as $r \to 0^+$, we obtain:

**Corollary 4.7** (No Interference). *Suppose $\psi(x) = \mathbb{1}(x = 0)$. Then, there exists $\ell, r$ s.t. the regret $R$ of the EXP3-IX-HT with the $(\ell, r)$-RRP satisfies*

$$R - R_{a^*} \lesssim \left(1 + \sqrt{\frac{1}{N} \log \frac{1}{\delta}}\right) \text{Reg}_{\text{OPT}} \qquad (1)$$

*w.p. $1 - \delta$ for every $a^* \in [k]$ and $\delta \in (0, \frac{1}{2})$.*

To better understand, recall that for adversarial bandits, the regret of EXP3-IX is $(1 + \sqrt{\log 1/\delta}) \cdot \text{Reg}_{\text{OPT}}$; see Chapter 12 in Lattimore and Szepesvári 2020. To see the difference, take $N = T$ and $\delta = N^{-\Omega(1)}$, this bound is $O(\log N \cdot \text{Reg}_{\text{OPT}})$, while (1) is $O(\text{Reg}_{\text{OPT}})$ for any $T$.

Another basic setting is $\kappa$-*neighborhood* interference (Leung 2022; Bojinov et al. 2023) where the reward ("potential outcome") of a unit depends only on the treatments of the units within distance $\kappa > 0$, i.e., $\psi(x) = \mathbb{1}(\kappa > x)$. In this case, by selecting $r = \kappa$, the $\psi$ terms become 0.

**Corollary 4.8** ($\kappa$-Neighborhood Interference). *Suppose $\psi(x) := \mathbb{1}(\kappa > x)$ for some $\kappa > 0$. Then, with $r = \kappa$ and $\ell = \kappa\sqrt{T}$, for any $a^* \in [k]$ and $\delta \in (0, \frac{1}{2})$, we have*

$$R - R_{a^*} \lesssim \left(1 + \kappa\sqrt{\frac{T}{N} \log \frac{1}{\delta}}\right) \text{Reg}_{\text{OPT}} \quad \text{w.p. } 1 - \delta.$$

Finally, consider the *power law $\psi(r)$*. This setting encompasses many fundamental settings, including the celebrated Cliff-Ord spatial autoregressive model (Cliff and Ord, 1973), where each unit's outcome is linear in its neighbors' treatments.

**Corollary 4.9** (Power-law Interference). *Suppose $\psi(r) = O(r^{-c})$ for a constant $c \geq 1$. Consider an $(\ell, r)$-RRP with $m = \min\{(N/T)^{\frac{2+c}{3+c}}, N^{\frac{2c}{2c+1}} T^{-\frac{2c-1}{2c+1}}\}$ clusters (and hence $\ell = \sqrt{N/m}$) and $r = \ell/\sqrt{T}$. Then, for any $a^* \in [k]$ and $0 < \delta \leq \frac{1}{2}$, w.p. $1 - \delta$ we have*

$$R - R_{a^*} \lesssim \left(1 + \sqrt{\frac{\ell^2}{N} \log \frac{1}{\delta}}\right) \text{Reg}_{\text{OPT}}. \qquad (2)$$

For example, when $c = 2$, we have $m = (N/T)^{4/5}$, and

$$(2) = \widetilde{O}(\sqrt{kT}) + k\frac{T^{9/10}}{N^{2/5}}\sqrt{\log \frac{1}{\delta}}.$$

### 4.5. Discussion: Interpretation in VaR

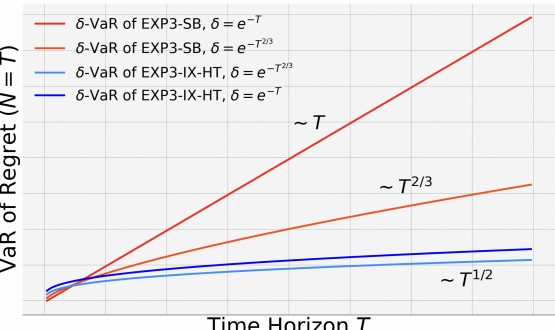

*Figure 2.* **VaR of Regret:** We visualize the $\delta$-VaR of regret for $\delta = e^{-T}$ and $\delta = e^{-T^{2/3}}$ respectively. Here we set $c = 1/2$. Our cluster-randomization based policy has a much lower VaR.

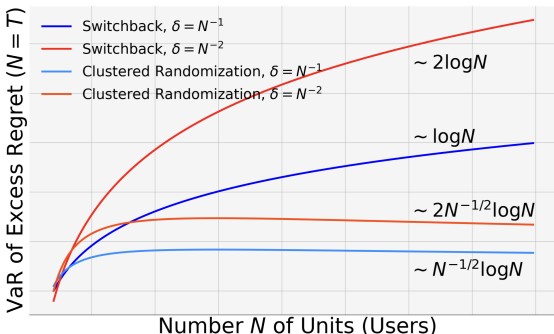

*Figure 3.* **VaR of Excess Regret:** The figure visualizes the excess regret $\text{Reg} - \text{Reg}_{\text{OPT}}$ that we can guarantee w.p. $1 - \delta$.

By taking $\eta \sim \sqrt{1/kT}$ in Theorem 4.6, our EXP3-IX-HT policy also achieves the optimal $\tilde{O}(\sqrt{kT})$ expected regret. However, our policy has a substantially lower tail risk compared to any switchback policy. To better illustrate, for concreteness, let us take $N = T$. Denote by $\text{Reg}_{\text{SB}}$ and $\text{Reg}_{\text{CR}}$ the regret (as random variables) of any SwitchBack policy and our EXP3-IX-HT policy (where "CR" means "Clustered Randomization"). Then, by Theorem 4.6,

$$\mathbb{P}\left(\text{Reg}_{\text{SB}} > \text{Reg}_{\text{OPT}} + \log \frac{1}{\delta}\right) \leq \delta \qquad (3)$$

and

$$\mathbb{P}\left(\text{Reg}_{\text{CR}} > \text{Reg}_{\text{OPT}} + N^{-c} \log \frac{1}{\delta}\right) \leq \delta \qquad (4)$$

for any $\delta > 0$, where $c > 0$ is a constant depending on $\psi$. To highlight the tail mass, we rewrite Equation (3) as

$$\mathbb{P}\left(\text{Reg}_{\text{SB}} - \text{Reg}_{\text{OPT}} > \tau\right) \leq e^{-\tau} \qquad (5)$$

and

$$\mathbb{P}\left(\text{Reg}_{\text{CR}} - \text{Reg}_{\text{OPT}} > \tau\right) \le e^{-N^c \tau} \qquad (6)$$

for any $\tau > 0$.

To see why $\text{Reg}_{\text{CR}}$ is more "robust", take $\tau = T^{2/3}$ and let $T$ range from 10 to 50. Then, the first probability in Equation (5) ranges from 2.7% to 0.01%, while the second is **astronomically** small. For example, if $N = T$ and $c = 1/2$, it ranges from $10^{-14}$ to $10^{-39}$.

More generally, consider $\delta = e^{-\alpha T}$ where $\alpha > 0$. Then,

$$V_{\text{SB}} = \text{Reg}_{\text{OPT}} + T^\alpha \text{ and } V_{\text{CR}} = \text{Reg}_{\text{OPT}} + \frac{T^\alpha}{\sqrt{N}} \sim \sqrt{T},$$

where "$\approx$" holds if $N \gg T$ (as in the real world). When $\alpha > 1/2$, the first bound is asymptotically larger.

**Acknowledgments.** We thank Qin Chao, Wang Chi Cheung, and Christina Lee Yu for their valuable feedback.

## Impact Statement

We introduce the Multi-Armed Bandits with Interference (MABI) framework and focus on cumulative reward maximization under spatial interference. Our work bridges adversarial bandits and causal inference, making it applicable to real-world systems such as online platforms and networked environments. Our cluster randomization-based policy achieves near-optimal expected regret with high-probability bounds that improve as the number of units increases. Unlike prior work, we do not rely on stationarity or sparse interference.

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

## A. Detailed Proof of Theorem 3.3

Recall that in the construction we chose $U = \{-2\sqrt{N}, 2\sqrt{N}\} \times \{-2\sqrt{N}, 2\sqrt{N}\}$. By re-scaling, w.l.o.g, let us assume that $\psi(0) = 1$ and $\psi(\sqrt{N}) = 0$. Consider the interior units

$$U_{\text{int}} := \{-\sqrt{N}, \sqrt{N}\} \times \{-\sqrt{N}, \sqrt{N}\}.$$

For each $u \in [N] \backslash U_{\text{int}}$, we define $Y_{ut} \equiv 0$ for all $t \in [T]$.

To define the reward function for the interior units, for each $u \in [N]_{\text{int}}$ we apply Lemma 3.4 with $m = \sqrt{N}$ and

$$V = \{u_x - m, u_x + m\} \times \{u_y - m, u_y + m\},$$

where $u = (u_x, u_y)$. Denote by $\tilde{f}_u : \{0, 1\}^V \to \mathbb{R}_+$ the function constructed in Lemma 3.4, and extend $f_u$ to $\{0, 1\}^N$ so that the function value of $u'$ is solely determined by its restriction on $V$, formally,

$$f_u(u') = \tilde{f}_u(u'|_V), \quad \forall u' \in \{0, 1\}^N.$$

By Lemma 3.4, $\tilde{f}_u$ satisfies the $\psi$-DIP, and so $f_u$ also satisfies the $\psi$-DIP.

Now, construct the reward function $Y_{ut}$. Consider i.i.d. variables $\xi_t \sim \text{Ber}(1/2)$ where $t \in [T]$. Define

$$Y_{ut}(z) = \frac{1}{2} + \left(\xi_t - \frac{1}{2}\right) f_u(z), \quad \forall z \in \{0, 1\}^N.$$

Then, by Lemma 3.4, $Y_{ut}(\cdot)$ satisfies the $\psi$-DIP. Moreover, for any fixed $z \in \{0, 1\}^N$, we have $\mathbb{E}_\xi[Y_{ut}(z)] = \frac{1}{2}$. Therefore, the expected reward of any policy is $T/2$.

Next, to show that the best fixed-arm policy has total expected reward $T/2 + \Omega(\sqrt{T})$, we need:

**Lemma A.1** (Bernoulli Anti-concentration Bound). *Let $(\xi_t)_{t \in [T]}$ be i.i.d. Bernoulli variables with mean $\frac{1}{2}$, and write $\xi = \sum_{t=1}^T \xi_t$. Then, for any $s \in [0, \frac{T}{8}]$, we have*

$$\mathbb{P}\left[\xi \geq \frac{T}{2} + s\right] \geq \frac{1}{15} \exp\left(-\frac{16s^2}{T}\right)$$

To conclude, let $R_a$ be the total reward of arm $a$. Consider the event $\mathcal{E} = \{R_1 \geq \frac{T}{2} + \frac{\sqrt{T}}{4}\}$. Since $R_1$ is the sum of $T$ i.i.d. Bernoulli's, by taking $s = \frac{1}{4}\sqrt{T}$ in Lemma A.1, we have $\mathbb{P}[\mathcal{E}] \geq \frac{1}{15}$. Therefore,

$$\mathbb{E}\left[\max\{R_0, R_1\} - \frac{T}{2}\right] \geq \mathbb{E}\left[\max\{R_0, R_1\} - \frac{T}{2} \middle| \bar{\mathcal{E}}\right] \cdot \mathbb{P}[\bar{\mathcal{E}}]$$

$$\geq \frac{1}{15} \cdot \frac{\sqrt{T}}{4} = \frac{\sqrt{T}}{60},$$

where the first inequality follows since $\max\{R_0, R_1\} \geq \frac{T}{2}$ a.s. $\qquad\square$

## B. Proof of Claim 3.5

Consider two cases. If $r_\star = r'_\star$, then the claim trivially follows from the definition of $\rho$. Now suppose that $r'_\star > r_\star$. Note that $r_\star, r'_\star$ are both integers, so $r'_\star \geq r_\star + 1$. By the definition of $r_\star$, there exists $v \in B(O, r_\star + 1) \backslash B(O, r_\star)$ s.t. $z_v = 0$ and $z'_v = 1$. Therefore, $z$ and $z'$ do not "agree" on $B(O, r_\star + 1)$, so

$$\rho < r_\star + 1. \tag{7}$$

On the other hand, since we assumed $r_\star < r'_\star$, we have $z_{B(O, r_\star)} = z'_{B(O, r_\star)}$, and thus

$$\rho \geq r_\star. \tag{8}$$

Combining Equations (7) and (8), we have $\rho = r_\star$. The claim follows by combining the two cases. $\qquad\square$

# C. Proof of Theorem 4.6

Recall that the HT-IX estimator is

$$\widehat{Y}_t(a) := \frac{1}{N} \sum_{u \in [N]} \frac{\mathbf{1}(X_{uta}^r(Z_t) = 1)}{Q_{uta}^r + \beta} Y_{ut}(Z_t).$$

We first decompose the regret using the following fake reward.

**Definition C.1** (Fake Reward). For each arm $a \in [k]$, we define

$$\widehat{R}_a = \sum_{t \in [T]} \widehat{Y}_t(a) \quad \text{and} \quad \widehat{R} = \sum_{t,a} P_{ta} \widehat{Y}_t(a).$$

In words, $\widehat{R}_a$ is approximately the total reward of always choosing arm $a$, where the true reward $\bar{Y}_t(a) = \frac{1}{N} \sum_{u \in [N]} Y_{ut}(a \cdot \mathbf{1}^N)$ is replaced with the HT-IX estimator $\widehat{Y}_t(a)$. Similarly, $\widehat{R}$ is a approximately the total reward of our policy. This is because the expected reward of our policy in round $t$ is approximately $\sum_a P_{ta} \bar{Y}_t(a)$, and $\widehat{Y}_t(a)$ is close to $\bar{Y}_t(a)$ since it is a good estimator.

Let us decompose the regret using the fake rewards. Recall that $R$ is the total reward of our EXP3-IX-HT policy, and that for any $a^* \in [k]$, $R_{a^*}$ is the reward of the fixed-arm policy at $a^*$. Then,

$$R - R_{a^*} = (R - \widehat{R}) + (\widehat{R} - \widehat{R}_{a^*}) + (\widehat{R}_{a^*} - R_{a^*}). \tag{9}$$

Next, we bound each of these three terms in a subsection separately.

## C.1. Bounding the First Term $R - \widehat{R}$

**Lemma C.2** (Bounding $R - \widehat{R}$). *It holds that* $R - \widehat{R} \le \frac{4rT}{\ell} + \beta \sum_{a \in [k]} \widehat{R}_a$.

*Proof.* We begin by further decomposing $R$ and $\widehat{R}$, allowing us to compare each term individually in the subsequent analysis. Let us write

$$R = \sum_{t=1}^T R_t \quad \text{where} \quad R_t = \frac{1}{N} \sum_{u \in [N]} Y_{ut}(Z_t), \quad \text{and} \quad \widehat{R} = \sum_{t=1}^T \widehat{R}_t \quad \text{where} \quad \widehat{R}_t = \sum_{a \in [k]} P_{ta} \widehat{Y}_t(a).$$

Now, fix any $t \in [T]$. Then,

$$\begin{aligned}
&N(R_t - \widehat{R}_t) \\
&= \sum_u Y_{ut}(Z_t) - \sum_u \sum_a \frac{P_{ta} \mathbf{1}(X_{uta}^r = 1)}{Q_{uta}^r + \beta} Y_{ut}(Z_t) \\
&= \sum_u \left( \mathbf{1}(X_{uta}^r = 0 \,\forall a \in [k]) + \sum_a \mathbf{1}(X_{uta}^r = 1) \right) Y_{ut}(Z_t) - \sum_u \sum_a \frac{P_{ta} \mathbf{1}(X_{uta}^r = 1)}{Q_{uta}^r + \beta} Y_{ut}(Z_t) \\
&\le \sum_u \mathbf{1}(X_{uta}^r = 0 \,\forall a \in [k]) + \sum_{u,a} \mathbf{1}(X_{uta}^r = 1) Y_{ut}(Z_t) - \sum_{u,a} \frac{P_{ta} \mathbf{1}(X_{uta}^r = 1) Y_{ut}(Z_t)}{Q_{uta}^r + \beta}, \tag{10}
\end{aligned}$$

where the inequality is because $Y_{ut}(\cdot) \le 1$. To proceeds, we make two observations. First, for any $u \in [N]$, the exposure mappings $X_{uta}^r$ can be all 0 **only** when $u$ lies close to $\partial C[u]$, the boundary of the cluster that contains $u$. Formally,

$$\mathbf{1}(X_{uta}^r = 0 \,\forall a \in [k]) \le \mathbf{1}(d(u, \partial C[u]) \le r).$$

Second, we note that

$$Q_{uta}^r \le \mathbb{P}\left[ Z_{C[u],t} = a \right] = P_{ta}.$$

Combining, we obtain

$$(10) \leq \mathbf{1}\left(d(u, \partial C[u]) \leq r\right) + \sum_{u,a} \mathbf{1}(X_{uta}^r = 1)Y_{ut}(Z_t) - \sum_{u,a} \frac{P_{ta} \cdot \mathbf{1}(X_{uta}^r = 1)}{P_{ta} + \beta} Y_{ut}(Z_t). \tag{11}$$

Note that in each cluster, there are at most $4r\ell$ units within a distance of $r$ to $\partial C[u]$. Since there are $N/\ell^2$ clusters, we have

$$\sum_{u \in [N]} \mathbf{1}(d(v, \partial C[u]) \leq r) \leq 4r\ell \cdot \frac{N}{\ell^2} = \frac{4rN}{\ell}.$$

It follows that

$$(11) \leq \frac{4rN}{\ell} + \sum_{u,a} \frac{\beta}{P_{ta} + \beta} \mathbf{1}(X_{uta}^r = 1)Y_{ut}(Z_t)$$

$$\leq \frac{4rN}{\ell} + \beta N \sum_{a \in [k]} \left( \frac{1}{N} \sum_{u \in [N]} \frac{\mathbf{1}(X_{uta}^r = 1)}{Q_{uta}^r + \beta} Y_{ut}(Z_t) \right)$$

$$= \frac{4rN}{\ell} + \beta N \sum_{a \in [k]} \widehat{Y}_t(a),$$

where inequalities follows again from $Q_{uta}^r \leq P_{ta}$. Therefore,

$$N\left(R - \widehat{R}\right) = N \sum_{t=1}^T \left(R_t - \widehat{R}_t\right) \leq \frac{4rNT}{\ell} + N\beta \sum_{a \in [k]} \widehat{R}_a,$$

and the lemma follows by dividing both sides by $N$. $\qquad\square$

### C.2. Bounding the Second Term $\widehat{R} - \widehat{R}_{a^*}$

We first derive an upper bound on $\widehat{R} - \widehat{R}_{a^*}$ by following the analysis of the EXP3 policy. The proof of the following can be found in the analysis of the EXP3 policy; see Equation (11.13) of (Lattimore and Szepesvári, 2020).

**Lemma C.3** (EXP3-style analysis). *Fix any $\eta \in (0, 1)$. Then, for any $a^* \in [k]$,*

$$\widehat{R} - \widehat{R}_{a^*} \leq \frac{\log k}{\eta} + \eta \sum_{t,a} P_{ta} \widehat{Y}_t(a)^2 \quad a.s.$$

We next show that $\widehat{Y}_t(a)$ is highly concentrated around $\bar{Y}_t(a)$, with a tail mass that vanishes in $N$. This is done by a careful analysis based on the Chernoff and Bernstein inequalities. We first introduce some basic tools.

**Theorem C.4** (Bernstein Inequality for i.i.d. Sum). *Let $\{X_i\}_{i=1,\ldots,n}$ be independent mean-zero random variables with $|X_i| \leq M$ a.s. where $M > 0$ is a constant. Then, for any $t > 0$,*

$$\mathbb{P}\left[\sum_{i=1}^n X_i \geq t\right] \leq \exp\left(-\frac{t^2}{\sum_{i=1}^n \mathbb{E}[X_i^2] + \frac{1}{3}Mt}\right).$$

**Theorem C.5** (Chernoff Inequality for i.i.d. Sum of Bernoulli's). *Suppose $X_1, \ldots, X_n \sim \mathrm{Ber}(p)$ are i.i.d. random variables and $\bar{\xi} = \frac{1}{n}\sum_{i=1}^n X_i$. Then, for any $\varepsilon > 0$, we have*

$$\mathbb{P}\left[\bar{\xi} \geq (1 + \varepsilon)\right] \leq \exp\left(-\frac{1}{3}\varepsilon^2 np\right).$$

**Lemma C.6** (Deviation of Bernoulli Sum). *Suppose $\delta, p \in (0, 1)$ and $X_1, \ldots, X_n$ are i.i.d. $\mathrm{Ber}(p)$ random variables. (1) Suppose $p \geq \frac{1}{n}$. Then with probability $1 - \delta$, we have*

$$\frac{1}{n} \sum_{i=1}^n X_i \leq p + \sqrt{\frac{p \log \frac{1}{\delta}}{n}}.$$

*(2) Suppose $p < \frac{1}{n}$, then w.p. $1 - \delta$,*

$$\frac{1}{n}\sum_{i=1}^{n} X_i \lesssim \sqrt{\log \frac{1}{\delta}} \cdot \frac{1}{n}.$$

*Proof.* **Part 1:** Suppose $p \geq \frac{1}{n}$. We will apply Bernstein's inequality on $(X_i - p)$. Since $X_i \in [0, 1]$ a.s. and $\mathbb{E}[(X_i - p)^2] = p(1-p) \leq p$, by taking $M = 1$ in Theorem C.4, we have

$$\mathbb{P}\left[\sum_{i=1}^{n} X_i \geq np + t\right] \leq \exp\left(-\frac{t^2}{\sum_{i=1}^{n}\mathbb{E}[(X_i - p)^2] + \frac{1}{3}t}\right) \leq \exp\left(-\frac{t^2}{np + \frac{t}{3}}\right). \tag{12}$$

for any $t > 0$. Let us choose $t = \sqrt{2np \log \frac{1}{\delta}}$. Since $p \geq \frac{1}{n}$, we have $np > t$. It follows that

$$(12) \leq \exp\left(-\frac{t^2}{2np}\right) \leq \delta.$$

**Part 2:** Suppose $p < \frac{1}{n}$. It suffices to consider i.i.d. $\widetilde{X}_i \sim \mathrm{Ber}(1/n)$ since $X_i$ stochastically dominates $X_i$. By the Chernoff bound (Theorem C.5), for any $\varepsilon > 0$,

$$\mathbb{P}\left[\frac{1}{n}\sum_{i=1}^{n}\tilde{X}_i \geq (1 + \varepsilon) \cdot \frac{1}{n}\right] \leq \exp\left(-\frac{1}{3}\varepsilon^2 n \cdot \frac{1}{n}\right).$$

In particular, for $\varepsilon = \sqrt{3 \log \frac{1}{\delta}}$, the above bound becomes $\delta$. $\qquad\square$

Next, we combine the above and bound $\sum_{t,a} P_{ta}\widehat{Y}_t(a)^2$.

**Lemma C.7** (Bounding the Squared Terms). *For any $\delta \in (0, 1)$, we have*

$$\sum_{t,a} P_{ta}\widehat{Y}_t(a)^2 \leq 512\left(1 + \frac{k\ell^2}{N}\log\frac{1}{\delta} + \frac{k}{\beta}\psi(r)^2\right)T \quad \text{w.p. } 1 - \delta.$$

*Proof.* Denote by $m = N/\ell^2$ the number of clusters. By the definition of $\widehat{Y}_t(a)$, for any $t \in [T], a \in [k]$,

$$\widehat{Y}_t(a) = \frac{1}{N}\sum_{u\in[N]}\frac{\mathbf{1}(X_{uta}^r = 1)}{Q_{uta}^r + \beta}Y_{ut}(Z_t) = \frac{1}{m}\sum_{C\in\Pi}\left(\frac{1}{\ell^2}\sum_{u\in C}\frac{\mathbf{1}(X_{uta}^r = 1)Y_{ut}(Z_t)}{Q_{uta}^r + \beta}\right). \tag{13}$$

By the $\psi$-DIP, if $X_{uta}^r = 1$, all units in $B(u, r)$ are assigned $a$, so $|Y_{ut}(Z_t) - Y_{ut}(a \cdot \mathbf{1}^N)| \leq \psi(r)$. Thus,

$$\mathbf{1}(X_{uta}^r = 1) \cdot Y_{ut}(Z_t) \leq \mathbf{1}(X_{uta}^r(Z_t) = 1) \cdot \left(Y_{ut}(a \cdot \mathbf{1}^N) + \psi(r)\right).$$

It follows that

$$(13) \leq \frac{1}{m}\sum_{C\in\Pi}\left(\frac{1}{\ell^2}\sum_{u\in C}\frac{\mathbf{1}(X_{uta}^r = 1)(Y_{ut}(a \cdot \mathbf{1}^N) + \psi(r))}{Q_{uta}^r + \beta}\right). \tag{14}$$

Moreover, by Proposition 4.2, we have $Q_{uta}^r \geq \frac{1}{8}P_{ta}$, so

$$(14) \leq \frac{1}{m}\sum_{C\in\Pi}\left(\frac{1}{\ell^2}\sum_{u\in C}\frac{\mathbf{1}(X_{uta}^r = 1)(Y_{ut}(a \cdot \mathbf{1}^N) + \psi(r))}{\frac{1}{8}(P_{ta} + \beta)}\right)$$

$$\leq \frac{8}{P_{ta} + \beta}\left(\frac{1}{m}\sum_{C\in\Pi}\frac{1}{\ell^2}\sum_{u\in C}\mathbf{1}(X_{uta}^r = 1)\left(Y_{ut}(a \cdot \mathbf{1}^N) + \psi(r)\right)\right) \tag{15}$$

Note that the cardinality of each cluster satisfies $|C| \le 2\ell^2$, assuming that $r \le \ell/2$. So, writing

$$\bar{Y}_{Ct}(a \cdot \mathbf{1}^N) := \frac{1}{|C|} \sum_{u \in C} Y_{ut}(a \cdot \mathbf{1}^N),$$

we obtain

$$(15) \le \frac{8}{P_{ta} + \beta} \left( \frac{1}{m} \sum_{C \in \Pi} \frac{2}{|C|} \sum_{u \in C} \mathbf{1}(X_{uta}^r = 1) \left( Y_{ut}(a \cdot \mathbf{1}^N) + \psi(r) \right) \right)$$

$$\le \frac{16}{P_{ta} + \beta} \left( \psi(r) + \sum_{\kappa \in \{0,1,2\}^2} \frac{1}{m} \sum_{C: \chi(C) = \kappa} \mathbf{1}(Z_{Ct} = a) \cdot \bar{Y}_{Ct}(a \cdot \mathbf{1}^N) \right). \tag{16}$$

Note that for each color $\kappa$, the above sum involves $\frac{1}{9}m$ independent random variables. So by Lemma C.6, w.p. $1 - \delta$ it holds that

$$\frac{1}{\frac{1}{9}m} \sum_{C: \chi(C) = \kappa} \mathbf{1}(Z_{Ct} = a) \cdot \bar{Y}_{Ct}(a \cdot \mathbf{1}) \le P_{ta}\bar{Y}_t(a) + \sqrt{\frac{P_{ta}\bar{Y}_t(a) + \log\frac{1}{\delta}}{\frac{1}{9}m}}$$

Combined with Equation (16), we conclude that

$$\sum_{t,a} P_{ta}\widehat{Y}_t(a)^2 \le \sum_{t,a} P_{ta} \left( \frac{16}{P_{ta} + \beta} \left( P_{ta}\bar{Y}_t(a) + \sqrt{\frac{P_{ta}\bar{Y}_t(a) + \log\frac{1}{\delta}}{\frac{1}{9}m}} + \psi(r) \right) \right)^2$$

$$\le 256 \sum_{t,a} \frac{P_{ta}}{(P_{ta} + \beta)^2} \cdot 2 \left( P_{ta}^2 \bar{Y}_t(a)^2 + \frac{P_{ta}\bar{Y}_t(a) + \log\frac{1}{\delta}}{\frac{1}{9}m} + \psi(r)^2 \right)$$

$$\le 512 \left( \sum_{t,a} P_{ta} + 9 \sum_{t,a} \frac{\log\frac{1}{\delta}}{m} + \sum_{t,a} \frac{\psi(r)^2}{P_{ta} + \beta} \right)$$

$$\le 512 \left( T + 9\frac{k\ell^2 T}{N} \log\frac{1}{\delta} + \frac{kT}{\beta}\psi(r)^2 \right),$$

where the second inequality follows since for any $a, b, c \in \mathbb{R}$, we have $(a + b + c)^2 \le 2(a^2 + b^2 + c^2)$. □

### C.3. Bounding the Third Term $\widehat{R}_a - R_a$

We need the following tool, which is stated as Lemma 12.2 in Lattimore and Szepesvári 2020.

**Proposition C.8** (One-sided Cramer-Chernoff Bound). *Let $\beta > 0$ and $\mathbb{F} = (\mathcal{F}_t)_{t \in [T]}$ be a filtration. Let $\{y_{t\alpha}\}_{t \in [T], \alpha \in \mathcal{A}}$ be real numbers, where $\mathcal{A}$ is a finite set. Let $(\delta_{t\alpha})$ be an $\mathbb{F}$-predictable process and $(Y_{t\alpha})$ be an $\mathbb{F}$-adapted process. Suppose for each $t \in [T]$, $Y_{t\alpha}$ is*
*(i) unbiased: $\mathbb{E}[Y_{t\alpha}|\mathcal{F}_{t-1}] = y_{t\alpha}$ for each $\alpha \in \mathcal{A}$,*
*(ii) negative correlated: for any $S \subseteq \mathcal{A}$ with $|S| >> 1$, we have*

$$\mathbb{E}\left[ \prod_{\alpha \in S} Y_{t\alpha} \,\middle|\, \mathcal{F}_{t-1} \right] \le 0, \quad \text{and}$$

*(iii) reasonably bounded: $0 \le \beta Y_{t\alpha} \le 2\delta_{t\alpha}$ a.s. for any $\alpha \in \mathcal{A}$.*
*Then, for any $\delta \in (0, 1)$, we have*

$$\mathbb{P}\left[ \sum_{t \in [T]} \sum_{\alpha \in \mathcal{A}} \beta \left( \frac{Y_{t\alpha}}{1 + \delta_{t\alpha}} - y_{t\alpha} \right) \ge \log\frac{1}{\delta} \right] \le \delta.$$

*Proof.* It suffices to show that $M_n := \prod_{t=1}^n \xi_t$ is a super-martingale (indexed by $n$) where

$$\xi_t = \exp\left(\sum_{\alpha \in \mathcal{A}} \beta\left(\frac{Y_{t\alpha}}{1 + \delta_{t\alpha}} - y_{t\alpha}\right)\right).$$

In fact, if this is true, then by Markov's inequality,

$$\mathbb{P}\left[\prod_{t=1}^n \xi_t \geq \frac{1}{\delta}\right] \leq \mathbb{P}\left[M_n \geq \frac{1}{\delta}\mathbb{E}[M_n]\right] \leq \delta,$$

which completes the proof.

Now we show that $(M_n)$ is a super-martingale. We first use the Cramer-Chernoff method to prove that $\mathbb{E}_{t-1}[\xi_t] \leq 1$ for any $t \in [T]$. We will use the following fact: for any $x > 0$, we have

$$\exp\left(\frac{x}{1+\lambda}\right) \leq 1 + x \leq e^x. \tag{17}$$

Write $\mathbb{E}_t[\cdot] := \mathbb{E}[\cdot|\mathcal{F}_t]$ for each $t \in [T]$. Then,

$$
\begin{aligned}
\mathbb{E}_{t-1}\left[\exp\left(\sum_{\alpha \in \mathcal{A}} \frac{\beta Y_{t\alpha}}{1 + \delta_{t\alpha}}\right)\right] &= \mathbb{E}\left[\prod_{\alpha \in \mathcal{A}} \exp\left(\frac{\beta Y_{t\alpha}}{1 + \delta_{t\alpha}}\right)\right] \\
&= \mathbb{E}\left[\prod_{\alpha \in \mathcal{A}} (1 + \beta Y_{t\alpha})\right] &&\text{by the first inequality in Equation (17)} \\
&\leq \mathbb{E}\left[1 + \beta \sum_{\alpha \in \mathcal{A}} Y_{t\alpha}\right] &&\text{negative correlation} \\
&= 1 + \beta \sum_{\alpha \in \mathcal{A}} y_{t\alpha} &&\text{by unbiasedness} \\
&\leq \exp\left(\beta \sum_{\alpha \in \mathcal{A}} y_{t\alpha}\right) &&\text{by the second inequality in Equation (17).}
\end{aligned}
$$

Rearranging, we deduce that

$$\mathbb{E}_{t-1}[\xi_t] \leq 1 \tag{18}$$

Thus, by the tower rule, for any $n \geq 1$ we have

$$\mathbb{E}\left[\prod_{t=1}^n \xi_t\right] = \mathbb{E}\left[\mathbb{E}_{n-1}\left[\prod_{t=1}^n \xi_t\right]\right] = \mathbb{E}\left[\prod_{t=1}^n \xi_n \cdot \mathbb{E}_{n-1}[\xi_t]\right] \leq \mathbb{E}\left[\prod_{t=1}^{n-1} \xi_t\right].$$

By induction, we deduce that $\mathbb{E}\left[\prod_{t=1}^n \xi_t\right] \leq 1$, and hence $(M_n)$ is a super-martingale. $\square$

To proceed, let us apply the above with clusters as the index "$\alpha$". Specifically, for a fixed RRP $\Pi$, let us decompose $\widehat{Y}_t(a)$ as a sum over the clusters by writing

$$\widehat{Y}_t(a) = \frac{1}{N} \sum_{u \in [N]} \frac{\mathbf{1}(X_{uta}^r = 1)}{Q_{uta}^r + \beta} Y_{ut}(Z_t) = \frac{1}{N} \sum_{C \in \Pi} Y_{Cta}$$

where

$$Y_{Cta} := \sum_{u \in C} \frac{\mathbf{1}(X_{uta}^r = 1)}{Q_{uta}^r + \beta} Y_{ut}(Z_t).$$

However, the negative correlation condition (ii) does not hold in general. In fact, consider two neighboring clusters $C, C' \in \Pi$ and units $u \in C$ and $u' \in C'$ with $B(u, r) \cap B(u', r) \neq \emptyset$. Then, for any $a \in [k]$, the exposure mappings $X_{uta}^r$ and $X_{u'ta}^r$ are *positively* correlated and therefore $Y_{Cta}$ and $Y_{C'ta}$ are dependent.

We avoid this obstacle by coloring the clusters so that $Y_{Cta}$'s with the same color are independent.

**Definition C.9** (Nine-Coloring). Fix an $(\ell, r)$-RRP $\Pi$ of $[0, \sqrt{N}]^2$. A mapping $\chi : \Pi \to \{0, 1, 2\}^2$ is a valid 9-*coloring* if for any $(i, j), (i', j')$,

$$\chi(C_{ij}) = \chi(C_{i'j'}) \quad \Longleftrightarrow \quad i \equiv i' \text{ and } j \equiv j' \pmod 3.$$

We show that two clusters of the same color have independent contributions to $\widehat{Y}_t(a)$.

**Lemma C.10** (Independence within Color Class). *Let $\chi$ be a valid 9-coloring of $\Pi$, an $(\ell, r)$-RRP. Then, for any $C, C' \in \Pi$ with $\chi(c) = \chi(c')$, and $u \in C, u' \in C'$, we have $Y_{Cta} \perp Y_{C'ta}$ for all $a \in [k]$.*

To see this, note that each $Y_{Cta}$ is determined by the arm assigned to the clusters. Formally, conditional on the history up to the $(t-1)^{\text{st}}$ round, $Y_{Cta}$ only depends on $\{Z_{C',t} \mid C' \in \Gamma(C)\}$ where $\Gamma(C) = \{C' \cap B(u, r) \neq \emptyset \text{ for some } u \in C'\}$. Lemma C.10 then follows by noting that $Y_{Cta}$ and $Y_{C'ta}$ are determined by two *entirely* different sets of random variables, i.e., $\Gamma(C)' \cap \Gamma(C) = \emptyset$.

We use Lemma C.10 and Proposition C.8 to bound the deviation of $\widehat{R}_a$ from $R_a$. Recall that the number $m$ of clusters in an $(\ell, r)$-RRP for $[0, \sqrt{N}]^2$ satisfies $m\ell^2 = N$.

**Proposition C.11** (Bounding $\widehat{R}_a - R_a$). *It holds that*

$$\max_{a \in [k]} \left\{ \widehat{R}_a - R_a \right\} \le \frac{8}{\beta m} \log \frac{1}{\delta} + \frac{T}{\beta \ell^2} \psi(r) \quad \text{and} \quad \sum_{a \in [k]} \left( \widehat{R}_a - R_a \right) \le \frac{8}{\beta m} \log \frac{1}{\delta} + \frac{T}{\beta \ell^2} \psi(r).$$

*Proof.* Denote by $N_c$ the (random) number of units in each cluster $c$, then $N_c \le 2\ell^2$ since $r \le \ell/2$. Then,

$$
\begin{aligned}
m \left( \widehat{R}_a - R_a \right) &= \sum_{t=1}^{T} \sum_{C \in \Pi} \frac{1}{\ell^2} \sum_{u \in C} \left( \frac{\mathbf{1}(X_{uta}^r = 1) Y_{ut}(A_t)}{Q_{uta}^r + \beta} - Y_{ut}(a \cdot \mathbf{1}) \right) \\
&\le \sum_{t=1}^{T} \sum_{C \in \Pi} \frac{1}{\ell^2} \sum_{u \in C} \left( \frac{\mathbf{1}(X_{uta}^r = 1) \cdot (Y_{ut}(a \cdot \mathbf{1}) + \psi(r))}{Q_{uta}^r + \beta} - Y_{ut}(a \cdot \mathbf{1}) \right) \\
&= \sum_{t=1}^{T} \sum_{C \in \Pi} \frac{1}{\ell^2} \sum_{u \in C} \left( \frac{\mathbf{1}(X_{uta}^r = 1) Y_{ut}(a \cdot \mathbf{1})}{Q_{uta}^r + \beta} - Y_{ut}(a \cdot \mathbf{1}) \right) + \sum_{t=1}^{T} \sum_{C \in \Pi} \frac{1}{\ell^2} \sum_{C \in \Pi} \frac{\psi(r)}{Q_{uta}^r + \beta} \\
&\le \sum_{t=1}^{T} \sum_{C \in \Pi} \frac{2}{N_c} \sum_{u \in C} \left( \frac{\mathbf{1}(X_{uta}^r = 1)}{1 + \frac{\beta}{Q_{uta}^r}} Y_{ut}(a \cdot \mathbf{1}) - Y_{ut}(a \cdot \mathbf{1}) \right) + \frac{mT}{\beta \ell^2} \psi(r), \quad (19)
\end{aligned}
$$

where the first equality follows since $m\ell^2 = N$, and the final inequality is because $Q_{uta}^r \ge 0$ and $N_c \le 2\ell^2$ a.s. To apply Proposition C.8, consider an unbiased estimate

$$\widetilde{Y}_{ut}(a) := \frac{\mathbf{1}(X_{uta}^r = 1)}{Q_{uta}^r} Y_{ut}(a \cdot \mathbf{1}),$$

for $Y_{ut}(a \cdot \mathbf{1})$. Then,

$$
\begin{aligned}
(19) &= 2 \sum_{t=1}^{T} \sum_{C \in \Pi} \frac{1}{N_c} \sum_{u \in C} \left( \frac{1}{1 + \frac{\beta}{Q_{uta}^r}} \widetilde{Y}_{ut}(a) - Y_{ut}(a \cdot \mathbf{1}) \right) + \frac{mT}{\beta \ell^2} \psi(r) \\
&\le 2 \sum_{\kappa \in [4]} \sum_{t=1}^{T} \sum_{C:\chi(C)=\kappa} \left( \frac{1}{1 + \frac{\beta}{P_{ta}}} \frac{1}{\ell^2} \left( \sum_{u \in C} \widetilde{Y}_{ut}(a) \right) - \bar{Y}_{Ct}(a \cdot \mathbf{1}) \right) + \frac{mT}{\beta \ell^2} \psi(r),
\end{aligned}
$$

where the inequality is because $Q_{uta}^r \le P_{ta}$.

We conclude by applying the Cramer-Chernoff inequality to the above. In Proposition C.8, take

$$Y_{t\alpha} := \frac{1}{N_c} \sum_{u \in C} \widetilde{Y}_{ut}(a) \quad \text{and} \quad \delta_{t\alpha} := \frac{\beta}{P_{ta}}.$$

Then, condition (i) in Proposition C.8 is satisfied since

$$\mathbb{E}[Y_{t\alpha}] = \mathbb{E}\left[\frac{1}{N_c}\sum_{u\in C}\widetilde{Y}_{ut}(a)\right] = \bar{Y}_{ct}(a\cdot\mathbf{1}).$$

Moreover, for each color $\kappa$, each term in the summation $\sum_{c:\chi(c)=\kappa}$ are independent, so (ii) is satisfied. Finally, note that for each $t, a$ we have

$$\beta Y_{t\alpha} = \beta\frac{1}{N_c}\sum_{u\in C}\widetilde{Y}_{ut}(a) = \beta\frac{1}{N_c}\sum_{u\in C}\frac{\mathbf{1}(X_{uta}^r = 1)}{Q_{uta}^r}Y_{ut}(a\cdot\mathbf{1}) \leq \frac{8\beta}{P_{ta}},$$

where the last inequality follows since $Q_{uta}^r \geq P_{ta}/8$ and $0 \leq Y_{ut}(\cdot) \leq 1$. Therefore, by Proposition C.8, we conclude that

$$\beta m\left(\widehat{R}_a - R_a\right) \leq 2\beta\sum_{\kappa\in[4]}\sum_{t=1}^T\sum_{c:\chi(c)=\kappa}\left(\frac{1}{1+\frac{\beta}{P_{ta}}}\frac{1}{N_c}\left(\sum_{u\in C}\widetilde{Y}_{ut}(a)\right) - \bar{Y}_{ct}(a\cdot\mathbf{1})\right) + \frac{mT}{\ell^2}\psi(r)$$

$$\leq 8\log\frac{1}{\delta} + \frac{mT}{\ell^2}\psi(r),$$

i.e.,

$$\widehat{R}_a - R_a \leq \frac{8}{\beta m}\log\frac{1}{\delta} + \frac{T}{\beta\ell^2}\psi(r).$$

The proof of (2) is identical by replacing every "$\sum_{u,t}$" with "$\sum_{u,t,a}$". $\qquad\square$

### C.4. Proof of Theorem 4.6

We begin by recalling the regret (w.r.t. a fixed arm $a^*$) decomposition:

$$R - R_{a^*} = (R - \widehat{R}) + (\widehat{R} - \widehat{R}_{a^*}) + (\widehat{R}_{a^*} - R_{a^*}).$$

Let us bound each term separately using the lemmas we have shown so far. By Lemma C.2,

$$R - \widehat{R} \lesssim \frac{rT}{\ell} + \beta\sum_{a\in[k]}\widehat{R}_a$$

$$= \frac{rT}{\ell} + \beta\sum_{a\in[k]}R_a + \beta\sum_{a\in[k]}(\widehat{R}_a - R_a)$$

$$\leq \frac{rT}{\ell} + \beta kT + \beta\sum_{a\in[k]}(\widehat{R}_a - R_a). \tag{20}$$

By Lemmas C.3 and C.7, w.p. $1 - \delta$ we have

$$\widehat{R} - \widehat{R}_{a^*} \leq \frac{\log k}{\eta} + \eta\sum_{t,a}P_{ta}\widehat{Y}_t(a)^2$$

$$\lesssim \frac{\log k}{\eta} + \eta T\left(1 + \frac{k}{m}\log\frac{1}{\delta} + \frac{k}{\beta}\psi(r)^2\right). \tag{21}$$

Combining Equations (20) and (21),

$$R - R_{a^*} = (R - \widehat{R}) + (\widehat{R} - \widehat{R}_{a^*}) + (\widehat{R}_{a^*} - R_{a^*})$$

$$\lesssim \left(\frac{rT}{\ell} + \beta kT + \beta\sum_{a\in[k]}(\widehat{R}_a - R_a)\right) + \left(\frac{\log k}{\eta} + \eta T\left(1 + \frac{k}{m}\log\frac{1}{\delta} + \frac{k}{\beta}\psi(r)^2\right)\right) + (\widehat{R}_{a^*} - R_{a^*})$$

$$\lesssim \beta kT + \frac{rT}{\ell} + \frac{\log k}{\eta} + \eta T\left(1 + \frac{k}{m}\log\frac{1}{\delta} + \frac{k}{\beta}\psi(r)^2\right) + \frac{1}{\beta m}\log\frac{1}{\delta} + \frac{T}{\beta\ell^2}\psi(r), \tag{22}$$

where the last inequality follows from Proposition C.11 and that $\beta \leq 1$. The statement follows by rearranging the terms into three categories: (i) those that involve $\log\frac{1}{\delta}$, (ii) those that do not involve $\log\frac{1}{\delta}$ but involve $\eta$, and (iii) others. $\qquad\square$

## D. Proof of Proposition 4.2

Consider two cases.

**Case 1.** Suppose $B(u, r)$ intersects some interior, say $I_{ij}$. Then, $B(r, r)$ intersects at most 2 strips and 1 quad. The probability that these 3 regions are assigned to $C_{ij}$ is

$$\frac{1}{2} \times \frac{1}{2} \times \frac{1}{4} = \frac{1}{16}.$$

**Case 2.** Suppose $B(u, r)$ intersects no interior or strips. Then, $B(u, r) \subseteq Q$ for some quad $Q$ and hence $\mathbb{P}[B(u, r) \subseteq C[u]] = 1$.

**Case 3.** Suppose $B(u, r)$ intersects no interior and exactly 1 strip $S$. Then, it must also intersect some quad $Q$. Denote by $I_{\alpha_1}, I_{\alpha_2}$ the two interiors that $S$ neighbors where $\alpha_1, \alpha_2 \in \{1, \ldots, B/\ell\}^2$. Then,

$$\mathbb{P}[B(u, r) \subseteq C[u]] = \sum_{i=1,2} \mathbb{P}[S \subseteq C_{\alpha_i}] \cdot \mathbb{P}[Q \subseteq C_{\alpha_i} \mid S \subseteq C_{\alpha_i}]$$

$$= \frac{1}{2} \cdot \frac{1}{4} + \frac{1}{2} \cdot \frac{1}{4} = \frac{1}{4}.$$

**Case 4.** Suppose $B(u, r)$ does not intersect any interior, and intersects 2 strips, denoted $S, S'$, and 1 quad, denoted $Q$. Then, $S, S'$ are neighboring the same interior, say $C_{ij}$. Then,

$$\mathbb{P}[B(u, r) \subseteq C_{ij}] = \mathbb{P}[S \subseteq C_{ij}] \cdot \mathbb{P}[S' \subseteq C_{ij}] \cdot \mathbb{P}[Q \subseteq C_{ij}] = \frac{1}{16}.$$

The statement follows by combining the above four cases. $\qquad\square$

## E. Experiments

We consider a 2-armed setting with $N$ units lying on a $\sqrt{N} \times \sqrt{N}$ lattice. We generate unit-level interference as follows. Each unit $u \in [N]$ is assigned a random reward $R_{ut}$. Let $\rho_{ut}$ be the proportion of the five immediate neighbors (counting $u$ itself) assigned arm 1 at time $t$. Then, the reward at $u$ is $(2\rho_{ut} - 1)c_{ut}$.

In two sets of experiments, we assume that $N = T^2$ and $N = T^3$ respectively, and let $T$ range from $10, 20, \ldots, 50$. For each fixed $N, T$, we randomly generate 100 instances. To necessitate exploration, we add large-scale non-stationarity by randomly generating drifts. Each drift is an 8-piecewise constant function, where the value on each piece is independently drawn from $U(0, 1)$. To align with the theoretical analysis, we partition the lattice into square-shaped clusters of side length $N^{-1/4}$. For simplicity, we perform a simplified version of the clustering without randomly assigning the boundary units to nearby clusters.

We compare the switchback version of EXP3-IX (denoted SB) and our clustered randomization-based policy, dubbed EXP3-IX-HT, and denoted "CR" in the figures. We evaluated the performance of the two policies by running each of them 200 times for each instance. We then compute the mean and the 95 percentile of the regret for each instance, and average these numbers over the 100 random instances.

We visualize the results in Figures 4 and 5. Consistent with the theoretical analysis, CR outperforms SB in terms of 95% percentile regret, without sacrificing mean regret. Moreover, we observe that when $N = T^3$, the regret exhibits a smaller deviation. Finally, we observe that when $N = T^3$, the 95 regret percentile of CR is lower compared to the $N = T^2$ case. This is reasonable since a larger $N$ helps reduce the variance in the reward estimation.

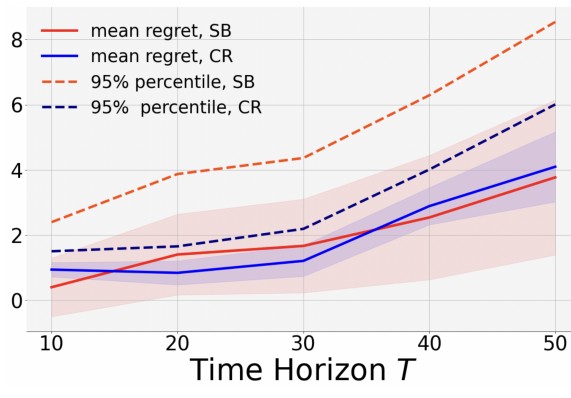
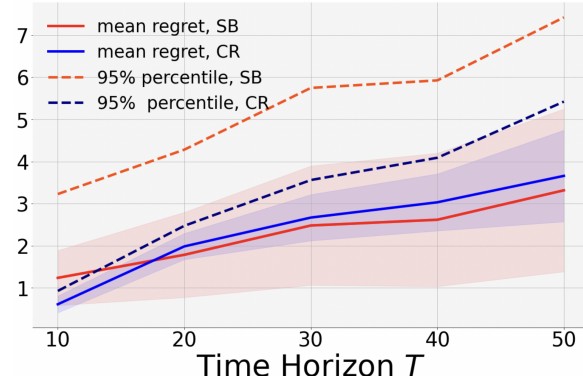

Figure 4. $N = T^2$ case

Figure 5. $N = T^3$ case

