# OpenReview forum: "Multi-Armed Bandits with Interference: Bridging Causal Inference and Adversarial Bandits"
_ICML.cc/2025/Conference — ICML 2025 poster_

### Official Review · Reviewer_SQ3C · 2025-02-18

**Overall Recommendation:** 3

**Summary:**

This paper is the first to study MAB with interference. The learning model in this paper requires each node to take the same action and assumes that the interference intensity decays with distance. The paper proposes an EXP3-IX algorithm based on exposure mapping, achieving a high-probability regret upper bound.

**Claims And Evidence:**

Yes

**Essential References Not Discussed:**

The vast majority of relevant references have been discussed.

**Experimental Designs Or Analyses:**

Yes

**Methods And Evaluation Criteria:**

Yes

**Other Comments Or Suggestions:**

## update after rebuttal: The authors have addressed my concerns, so I recommend acceptance.

**Other Strengths And Weaknesses:**

**Strengths:**
1. This paper proposed the first study on MAB with interference.
2. This paper proposes an algorithm that integrates exposure mapping. By leveraging exposure mapping and the assumption on interference intensity, this algorithm optimizes the performance of the general switchback strategy combined with EXP3-IX.
3. The experimental section of the paper is relatively comprehensive.

**Weaknesses/Problems:**
1. The regret definition in the paper only considers the optimal switchback policy, where each node takes the same action. This somewhat reduces the problem's difficulty, and the authors do not provide a clear motivating example from real-world applications.
2. The subheadings in the paper are somewhat confusing. Based on my understanding, a simple combination of EXP3-IX and the switchback policy is already sufficient to achieve a high-probability regret upper bound. I am not entirely sure why the authors named Section 3 "Expected Regret"; it might be because this chapter primarily establishes an expected regret lower bound. As for Section 4, I find "High Probability Regret" is also inappropriate, as obtaining a high-probability regret bound is fairly straightforward. From my perspective, the main contribution of the paper is integrating the exposure mapping mechanism into EXP3-IX and leveraging the assumption of interference attenuation to derive a regret bound that is tighter than the one obtained by directly applying EXP3-IX with the switchback policy.
3. The discussion of stochastic and adversarial settings in the related work section is somewhat controversial. In the stochastic case, it is typically assumed that the true reward is perturbed by a 1-sub-Gaussian noise, making the reward unbounded. In contrast, the adversarial case usually assumes that the reward is bounded. Given the setting of this paper, I believe the authors should state that their work and the referenced papers consider different settings, rather than claiming that the adversarial setting is more general than the stochastic one.
4. The authors do not provide motivated examples/references for the proposed Decaying Interference Property assumption.

**Questions For Authors:**

See the Strengths And Weaknesses section

**Relation To Broader Scientific Literature:**

N/A

**Theoretical Claims:**

I have checked most of the proofs and found that they are correct.

---

> ### Author Rebuttal · Authors · 2025-03-28
>
> Thank you for your insightful comments. We appreciate the feedback and would like to share our responses below.
>
> *1. Definition of Regret:*
>
> Alternatively, one could define the benchmark to be the optimal "personalized" treatment assignment from a function class $\mathcal{F}$. In this case, our analysis still applies—for instance, the expected regret bound scales with a factor of $\sqrt{|\mathcal{F}|}.$ We note that this "personalized" benchmark was used in Agarwal et al 2024, however, the cost paid for choosing this stronger benchmark is that the environment was assumed to be **stationary**.
>
> It’s also worth noting that the same critique could be made of many foundational works in causal inference, which focus on the ATE effect between all-1 and all-0 policies. While these benchmarks appear simplistic, they are far from trivial and have led to a rich body of work in causal inference. Of course, policy learning - finding a good mapping from covariates to treatment — is a natural extension. But it is viewed as a more advanced goal, and builds upon insights from the fixed-policy setting, analogous the one in this work.
>
> *2. Expected regret vs h.p. regret:*
>
> It is true that a h.p. bound can be obtained by treating the entire system as a single “unit” and applying h.p.bounds for EXP-IX (as we discussed at the end of the intro and the start of section 4). However, this approach completely ignores N, and as a result, the tail probability does not vanish with increasing N. This dependence on N is crucial, especially in practical applications where $N \gg T.$ In fact, we dedicate Section 4.5 specifically to discussing this distinction.
>
> From a practical standpoint, a “pure” switchback policy—where the entire system flips between treatment and control—is rarely used. Instead, tech firms typically partition the space into clusters and independently assign treatment or control to each cluster in each period. This is precisely what our algorithm is designed to capture. A concrete example is DoorDash’s use of “clustered” switchback experiments: https://careersatdoordash.com/blog/experiment-rigor-for-switchback-experiment-analysis/
> In this sense, a key contribution of our work is to lay the **theoretical foundation for a policy class that are widely deployed in industry.**
>
> While we mentioned the phrase “vanishing in N” in the abstract and other parts of the paper, we agree that this could have been more clearly emphasized. If the paper is accepted, we will revise accordingly to better highlight this point.
>
> *3. Discussion of stochastic and adversarial bandits in the literature review.*
>
> This is a fair point. To clarify, our claim that the adversarial setting is more general than the stochastic one assumes that the reward distribution has bounded support. That said, for the purposes of algorithm design and analysis, assuming sub-Gaussian noise with bounded mean behaves similarly to the bounded reward setting (e.g., Bernoulli rewards).
>
> *4.  Examples/references for the Decaying Interference Property*
>
> As we mentioned right before Defn 2.3, this Decaying Interference Property (DIP) is **borrowed from Leung 2022**. The best way to motivate the the DIP is by noting that it generalizes several natural interference models that are well-known in the literature, including:
> - (a) SUTVA (see our remark 2.4 right after def 2.3);
> - (b) k-neighborhood model, where two units interfere with each other if there distance is at most k;
> - (c) Cliff-Ord autoregressive model, where the treatment effect on unit u is a linear combination of the treatment effects on its direct neighbors (plus base effects). In this case, we have $\psi(r) = r^{-2}.$

---

> > ### Comment · Reviewer_SQ3C · 2025-04-05
> >
> > Thank you for your reply. The motivation regarding the switchback policy setting is no longer a concern. The authors may consider including these discussions in a future version of the paper.
> >
> > I still have some questions: When you mentioned "utilizing N," were you referring to the decay assumption? If this assumption is relaxed, do the properties discussed in the paper still hold? In addition, I find the assumption $N \gg T$ to be unrealistic in typical online settings. In most bandit tasks, the time horizon $T$ usually exceeds 10,000. If we assume $N \gg T$, say $N = 100000$, it would be extremely difficult to collect data from such a large number of units in real-time. Moreover, the case mentioned by the authors, where $\delta = e^{-\alpha T}$, is rarely considered in the bandit literature. In fact, most works assume a fixed confidence level with $\delta = \Theta(1)$, and the regret upper bound is typically analyzed as $\widetilde{O}(\sqrt{KT})$. Does the setting $\delta = e^{-\alpha T}$ have any particular significance in the context of MABI?
> >
> > Minors: Could the authors clarify the scale of the x-axis and y-axis in the experimental plots shown in Section 4.5?

---

> > > ### Author Response · Authors · 2025-04-06
> > >
> > > Q: *"When you mentioned "utilizing N," were you referring to the decay assumption?"*
> > >
> > > A: Not exactly. In fact, the decay function $\psi$ may not depend on N (e.g. in the spatial interference setting, \psi is a function of the max distance that two users can interfere).
> > >
> > > By "utilizing N", we meant that better statistical performance can be achieved  (e.g. for estimating treatment effects) when N is large; for example, many work in causal inference consider T=1, and the  variance of their estimators decrease in N. In contrast, switchback don't "utilize N" since they view the entire system as a single unit.
> > >
> > > Q: *"In addition, I find the assumption unrealistic in typical online settings.*
> > >
> > > A: good point. Here, a "round" can be interpreted as the amount of time we wait before we are allowed to change the treatments. In practice, its length can range from hours to months, and so T is typically hundreds/thousands; whereas N, the number of users, can be on the **millions**.
> > >
> > > Q: *"If we assume $N\gg T$, say $N=10^5$, it would be extremely difficult to collect data from such a large number of units in real-time."*
> > >
> > > A: True, but this actually **highlights** the advantage of our cluster-randomization approach - instead of collecting data from each user, all the policy needs is **cluster-level statistics** (e.g. mean revenue from a ZIP code region). Handling such data is much easier.
> > >
> > > Q: *"Moreover, the case mentioned by the authors, where $\delta = e^{-\alpha T}$, is rarely considered in the bandit literature. In fact, most works assume a fixed confidence level with $\delta=\Theta(1)$"*
> > >
> > > A: This is a good point. However, even when $\delta = \Theta(1)$, our approach still achieves an **asymptotically lower VaR**. To make this concrete, consider spatio-interference where two units interfere if their distance is at most $\kappa = O(1)$. By Corollary 4.8, the VaRs of our algorithm and the switchback policy scale as $\sqrt{ \frac{T}{N} \log \frac{1}{\delta} }$ and $\sqrt{ T \log \frac{1}{\delta} }$. In particular, if we take $N = T^2$ and $\delta = \Theta(1)$, then our algorithm’s VaR is $\sqrt{1/T}$, while the switchback policy’s VaR is $\sqrt{T}$; this is a sharp contrast as one is decreasing in T while the other is increasing.
> > >
> > > Q: *"Minors: Could the authors clarify the scale of the x-axis and y-axis in Section 4.5?"*
> > >
> > > A: in Fig 2, the x-axis is T, and the y-axis is the $(1-\delta)$-VaR of the regret, which is roughly $$E[{\rm regret}] +  O(\sqrt {\log
> > >  1/\delta})\cdot  {\rm Variance \ of\ regret}.$$
> > > In Fig 3, we consider an alternative perspective: the y-axis is the "excess regret", defined as $${\rm True\ regret} - {\rm Minimax\  optimal \ regret}.$$
> > > A **key distinction** is that the excess regret may be **decreasing** if we fix a relationship between $T,N$ such that $N\gg T$ (e.g. $N=T^2$)

---

### Official Review · Reviewer_XNnY · 2025-03-07

**Overall Recommendation:** 3

**Summary:**

The authors study a multi-armed bandit problem where interference (treatment of one arm affects the outcome of others) exists. The authors theoretically prove that a switchback policy achieves optimal regret. They provide a novel method based on clustered randomization and prove that the regret of the proposed method is both optimal in expectation and vanishing in N with high probability.

**Claims And Evidence:**

The authors concisely described key ideas of their proofs. They highlighted key steps and provided detailed arguments in the main paper.

**Essential References Not Discussed:**

I did not find any essential references not discussed.

**Experimental Designs Or Analyses:**

The results showcase the VaR of the regret. Although the experimental results are limited, I understand the focus of the paper is on theory and thus experiments are not the main contribution.

**Methods And Evaluation Criteria:**

Yes, the proposed method clearly addressed the shortcomings of existing methods.

To address the drawback of switchback policies that rates do not vanish as N grows, they propose a policy that has a regret bound vanishing in N. To achieve this, they integrate the truncated HT estimator into the EXP3-IX framework. There are two main challenges: the uniform spatial clustering is not robust since arms may have very low probabilities; it is unclear how to select the IX parameter in the batched setting.

**Other Comments Or Suggestions:**

How does the choice of the learning rate and the IX parameter beta affect the performance of the proposed method? Could the authors verify empirically that the proposed method is robust against choices of hyperparameters?

**Other Strengths And Weaknesses:**

The paper is well written and easy to follow.

**Questions For Authors:**

Please see my comments above.

**Relation To Broader Scientific Literature:**

The paper clearly discussed relevant literature.

**Theoretical Claims:**

The authors establish an upper bound restricted to switchback policies. They further establish a lower bound showing that rates cannot be improved by leveraging N using more complicated policies. The then prove the regret bound for their proposed method. I did not find any error in the theoretical claims.

---

> ### Author Rebuttal · Authors · 2025-04-01
>
> Thanks for your feedback.
>
> *"How does the choice of the learning rate and the IX parameter beta affect the performance of the proposed method?"*
>
> Good question. Intuitively, $\beta$ controls the forced exploration of arms with low score. This aligns very well with Theorem 4.6, where $\beta$ appears in both terms B and C. In fact, B corresponds to the tail mass, and a small $\beta$ (which means less exploration) would cause the tail mass to go up. On the other hand, C is the regret caused by the bias (due to both interference and bias in the estimator). If $\beta$ is away from the "sweet spot", the HT estimator has a high bias.
>
> Thanks again for this question, which helps improve the clarity of our results. If accepted, we will add these discussions to the paper.

---

### Official Review · Reviewer_MDpF · 2025-03-14

**Overall Recommendation:** 3

**Summary:**

The paper presents optimal expected regret bound and presents a high-probability regret bound in presence of correlated rewards among the arms.



post-rebuttal: I wish to thank the authors for the reply. I will keep my score.

**Claims And Evidence:**

Claims are well-supported by evidence.

**Essential References Not Discussed:**

Relevant literature is discussed.

**Experimental Designs Or Analyses:**

Experimental design and analysis make sense to me. Though the legends in benchmarks e.g. EXP3-SB is not well-explained -- specifically which method is this?

**Methods And Evaluation Criteria:**

I am unsure about how the distance-decaying interference assumption connects to other definitions of interference in the causal inference literature.

**Other Comments Or Suggestions:**

There seems to be a "depends" missing in the abstract "The reward of each unit on the treatments of all units, and this dependence decays in distance.'

**Other Strengths And Weaknesses:**

N/A

**Questions For Authors:**

How does the computational complexity analysis of the proposed algorithm look like?

**Relation To Broader Scientific Literature:**

It is not completely clear methodology-wise what is the essential difference between stochastic bandits that consider possibly more general notions of interference and the present paper that considers a specific notion of interference and adversarial reward.

**Theoretical Claims:**

I have not checked the correctness of any proofs for theoretical claims.

---

> ### Author Rebuttal · Authors · 2025-04-01
>
> Thanks for the feedback.
>
> Q: *"It is not completely clear methodology-wise what is the essential difference between stochastic bandits that consider possibly more general notions of interference and the present paper that considers a specific notion of interference and adversarial reward."*
>
> A: Good question. There are two disadvantages of the stochastic (i.i.d. reward) bandits analog:
> - Practicality: Real-world A/B testing involves highly dynamic and heterogeneous environments;
> - Technicality: If the rewards are iid, the problem wouldn't be too different Leung 2022. In fact, all we need to do is to combine their low-error estimator with a known paradigm for stochastic bandits, such as UCB.
>
> That said, we do agree that the inclusion of a dedicated section on the stochastic version would provide a more complete picture, and we will do so if accepted.
>
> Q: *"How does the computational complexity analysis of the proposed algorithm look like?"*
>
> A: The bottleneck in terms of the time complexity lies in computing the estimator. Fix an arm. For each unit $u$, we need to compute the exposure mapping probability. this involves examining the arm assigned to each of the $O((r/\ell)^2)$ clusters that are close to $u$. Therefore, the total run time per round is $O(kN (r/\ell)^2)$. (Note that $r/\ell\le \sqrt N$)

---

### Official Review · Reviewer_aFuM · 2025-03-14

**Overall Recommendation:** 3

**Summary:**

The authors combine Auer's EXP3 policy framework, the Horvitz-Thompson IPW (inverse propensity weighting) estimator, along with implicit exploration, and a clustered randomization scheme,
in order to achieve the optimal $O(\sqrt T)$ regret bound ($T$ being the horizon), while admitting a high-probability bound that vanishes with increasing $N$ (the number of experimal units),
under a spatially decaying interference property / assumption.

**Claims And Evidence:**

Theorems 3.3 on the regret for any MABI (including EXP3-based switchback) approach and Theorem 4.6 on the high-probability regret bound for the proposed Algorithm 1, supported via corollaries 4.7 - 4.9, appear to be the main theoretical claims in this paper.

The authors plot the VaR (Value at Risk ?) for their regret bounds against $T$ (the horizon) and $N$ (the number of experimal units), in Fig. 2 and Fig. 3, respectively, in order to demonstrate desirable properties of their regret bounds.

**Essential References Not Discussed:**

On the topic of cluster-based randomized experimentation, the authors may also wish to cite "Detecting Network Effects: Randomizing Over Randomized Experiments", by Saveski, et al.

**Experimental Designs Or Analyses:**

No experimental results are presented in this paper.

**Methods And Evaluation Criteria:**

The proposed method / algorithm (Algorithm 1) combines several known approaches mentioned under "Summary" in order to obtain novel theoretical results mentioned under "Claims and Evidence".
For a theoretical paper such as this one, validation via extensive proofs in the main body of the paper and the appendices appear to be appropriate evaluation criteria.

**Other Comments Or Suggestions:**

In the Impact statement on lin 419, "focus cumulative" should be "focus on cumulative".

**Other Strengths And Weaknesses:**

The paper is clearly written and the technical assumptions are clearly stated.
The robust random partition is illustrated quite well in Fig. 1.

**Questions For Authors:**

In the classic reference "Logarithmic regret algorithms for online convex optimization", by Hazan et al., it is shown how $O(\sqrt T)$ regret bounds can be converted into $O (log T)$ bounds via stronger assumptions. Have the authors considered a similar approach of using stronger assumptions ?

**Relation To Broader Scientific Literature:**

Please refer to the reference mentioned under "Questions For Authors".

**Theoretical Claims:**

Please refer to the theoretical claims mentioned under "Claims And Evidence".
This reviewer didn't check the proofs for Theorems 3.3 and 4.6 in detail.

---

> ### Author Rebuttal · Authors · 2025-04-01
>
> Thanks for your feedback!
>
> Q: *"In the classic reference 'Logarithmic regret algorithms for online convex optimization' by Hazan et al., it is shown how
>  regret bounds can be converted into ..."*
>
> A: This is an interesting thought. First, their result is not applicable to this work, since our reward functions are defined on a discrete domain and thus we can not define convexity (at least in the ordinary sense). But your intuition is right - if the reward function has a special form, such as linear in the neighbors treatment assignments (e.g. Cliff-Ord model) then there exists an estimator with lower error, and this would lead to better regret guarantees. This can be an interesting direction for future work.
>
> Q: *On the topic of cluster-based randomized experimentation, the authors may also wish to cite "On the topic of cluster-based randomized experimentation, the authors may also wish to cite "Detecting Network Effects: Randomizing Over Randomized Experiments", by Saveski, et al."*
>
> A: Yes, this is a relevant work; we will cite it in the updated version.

---

> > ### Comment · Reviewer_aFuM · 2025-04-08
> >
> > I wish to thank the authors for the response and the proposed revision. I will retain my original "weak accept" rating.

---

### Decision · Program_Chairs · 2025-05-01

**Decision:**

Accept (poster)

**Comment:**

The paper analyzes a novel framework—adversarial bandits with interference—and proposes a new algorithm that guarantees a tight upper bound on the expected regret and ensures that the tail mass of the regret vanishes as the number of treated units increases. The new algorithm integrates the clustered randomization idea (using a more sophisticated version-RRP) with the EXP3-IX algorithm, hence controlling the variance of the HT estimator for unobserved rewards.

Overall, all reviewers agree that the novel framework is well-motivated, and the proposed algorithm provides an effective solution with low variance. Authors are encouraged to incorporate reviewers’ comments (clarifying assumptions, adding references) in their revised version.